# FEDERATED LEARNING IN STREAMING SUBSPACE

## ABSTRACT

Federated learning (FL) has received widespread attention due to its distributed training and privacy protection. However, existing federated learning methods encounter significant challenges, such as increased communication costs and degraded model performance, when processing non-independently and identically distributed (non-IID) data. This paper jointly alleviates these problems by analyzing and exploiting the low-rank properties of global model trajectories.

Primarily, we introduce a streaming subspace update strategy and then propose a general federated learning framework, **F**erated **L**earning in **S**treaming **S**ubspace (`FLSS`). In `FLSS`, local model updates are restricted to the global streaming subspace, resulting in low-dimensional trajectories. The server then aggregates these trajectories to update the global model. Comprehensive experiments verify the effectiveness of our framework. In Cifar100, the `FLSS`-equipped FL method outperforms the baseline by 2.14% and reduces the communication cost by 80%. `FLSS` utilizes the early training information of the global model to simultaneously improve the performance and communication efficiency of federated learning.

## 1 INTRODUCTION

The remarkable progress made in machine learning is largely due to the availability of abundant and extensive data Cordts et al. (2016); Lin et al. (2014); Russakovsky et al. (2015). Nonetheless, as data volumes grow, aggregating such data becomes very difficult. Consequently, federated learning emerged as a distributed framework for machine learning Li et al. (2021b; 2020a); Yang et al. (2019). Federated learning leverages data stored on edge devices such as smartphones and PCs to collaboratively train global models under scheduling by a central server Kairouz et al. (2021); Li et al. (2020a); Mothukuri et al. (2021), which is of great significance in privacy-sensitive applications Bonawitz et al. (2019); Kaissis et al. (2020); Dayan et al. (2021); Hard et al. (2018).

Within the domain of federated learning, the communication cost of handling heterogeneous data is a significant obstacle Konecnỳ et al. (2016); Wu et al. (2022); Kairouz et al. (2021). This challenge stems mainly from the essential requirement of transmitting local models or gradients from each client to a central server Li et al. (2022a). Moreover, in the non-independently and identically distributed (non-IID) scenario, the imbalance of client data further affects the performance of the model Li et al. (2019); Hsu et al. (2019); Zhang et al. (2023b). Hence, two pivotal challenges emerge as focal points for research in federated learning: ❶ reducing communication overhead Konecnỳ et al. (2016) and ❷ mitigating the accuracy degradation problem resulting from data heterogeneity Konečnỳ et al. (2016).

Numerous studies have been conducted to address these challenges Zhu et al. (2021); Li et al. (2021a); Acar et al. (2021); Bernstein et al. (2018). In the context of addressing data heterogeneity, FL methods are primarily categorized into update correction Karimireddy et al. (2020), regularization Li et al. (2020b), model splitting Li et al. (2021a), and knowledge distillation Zhu et al. (2021). However, these methods often do not fully leverage early information from the global model. Regarding communication compression, approaches such as Fetchsgd Rothchild et al. (2020), Signsgd Bernstein et al. (2018); Karimireddy et al. (2019), and STC Sattler et al. (2019) introduce gradient compression techniques like sketching, quantization, and sparsification. Nevertheless, these compression frameworks are randomized and data-independent, which inherently limits their effectiveness. While existing algorithms are effective in tackling individual issues, they often struggle to simultaneously address both communication overhead and data heterogeneity challenges.

Therefore, we focus on designing a strategy that can handle these problems simultaneously, which can be applied to most federated learning frameworks, further improve its performance, and reduce communication costs. Inspired by the representational redundancy exhibited by neural networks Li et al. (2022b); Jacot et al. (2018); Gressmann et al. (2020), we performed principal component analysis on the training trajectories of the global model, as shown in Fig. 2. We find that a rough model update can be constructed with fewer basis vectors. Consequently, we propose to restrict local model update to a low-dimensional global subspace. This approach maximizes the utilization of global model information by aligning local updates within the global subspace, thereby enhancing overall performance. Additionally, communication costs are reduced by transferring the projection coefficients of the model within this subspace.

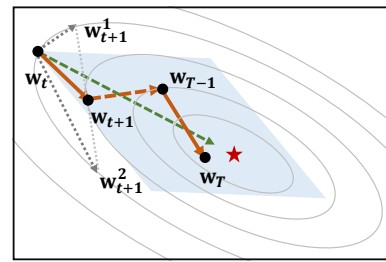

Figure 1: Model trajectory diagram of FedAvg. Its global model trajectory is mainly distributed in a low-dimensional subspace (blue diamond). The green dotted line is the main direction of this subspace.

Nonetheless, a key challenge in training federated learning models within subspaces lies in obtaining a subspace that maintains training performance. Existing subspace extraction solutions primarily involve random generation or pre-training sampling, which often struggle to cover the complete neural network parameter space. Applying it to federated learning may lead to a loss of accuracy. To address this issue, we design a strategy inspired by subspace tracking algorithms Eftekhari et al. (2019); Řehůřek (2011); Grammenos et al. (2020), which involves real-time monitoring of changes in neural network parameters to ensure that the subspace consistently contains the latest model information. Building upon this subspace, we propose a novel federated learning training scheme named *Ferated Learning in Streaming Subspace* (**FLSS**).

In FLSS, the training process involves constraining the local model to a low-dimensional subspace of the global model trajectory, which is equivalent to unifying the local model updates under the global subspace and performing local model fine-tuning, thus mitigating the harm of statistical heterogeneity. The consensus among the models is achieved by aggregating the projection coefficients of the local model update residing in this subspace, resulting in reduced communication costs. To cultivate the streaming subspace on the client side, a subspace tracking method is employed. This method periodically samples the global model trajectory and performs singular value decomposition to capture the knowledge of model changes during the training phase. Extensive experiments show that we obtain better performance by applying FLSS to the traditional FL algorithms. For instance, compared with traditional FedAvg, after applying the FLSS strategy, we obtained an accuracy improvement of $1.90 \sim 2.58\%$ and $0.08 \sim 1.76\%$ on Cifar10 and Cifar100 with different degrees of heterogeneity, respectively, and reduced the communication cost by nearly $80\%$. In summary, the key contributions of this paper can be summarized as follows:

- We propose FLSS, which limits local model updates to the streaming subspace of global model trajectory to fully exploit global information, and reduces communication costs by transmitting the projection coefficients of model updates in the subspace.

- We introduce a strategy to extract the streaming subspace for training. This strategy ensures that the subspace always contains the latest information of the model by performing singular value decomposition on the global model trajectory in real-time.

- Through multiple datasets, we verify that FLSS can improve the FL method in terms of communication efficiency and model performance. Specifically, with FedAvg, using FLSS reduced communication costs by $80\%$ and increased accuracy by up to $8.15\%$.

## 2 RELATED WORK

Due to space limitations, only most related works to this paper are discussed here. For more detailed illustrations, please refer to the Appendix A.

**Federated Learning.** FL algorithms that improve performance in heterogeneous scenarios mainly include four categories: regularization Li et al. (2020b); Acar et al. (2021); Kim et al. (2022), model splitting Li et al. (2021a); Jiang et al. (2022), knowledge distillation Zhu et al. (2021); Lee et al.

(2022); Gong et al. (2022); Huang et al. (2022), and update correction Karimireddy et al. (2020); Gao et al. (2022); Niu & Deng (2022). Specifically, FedProx Li et al. (2020b) adds a regularization term to local loss. MOON Li et al. (2021a) combines contrastive learning to align local and global features. FedGen Zhu et al. (2021) utilizes generators to ensemble local knowledge and guide local training, but it also brings non-negligible communication and computational overhead. SCAFFOLD Karimireddy et al. (2020) counters local model drift using gradient calibration. Although they address statistical heterogeneity, it does not reduce message size nor fully exploit early global model information.

For communication efficient FL methods, Fetchsgd Rothchild et al. (2020) uses sketches to compress local gradients. The Signsgd+EF Karimireddy et al. (2019) framework reduces communication costs and enhances the generalization of signsgd through 1-bit quantization and error feedback. Furthermore, STC Sattler et al. (2019) combines top-k sparsification with quantization. Although these methods can reduce communication costs, they are less effective in heterogeneous scenarios due to the randomness of compression and insufficient consideration of global information.

**Training in Tiny Subspace.** Many studies have emphasized the inherent low-dimensional characteristics of neural networks Tuddenham et al. (2020); Vinyals & Povey (2012); Gressmann et al. (2020). A seminal investigation in Li et al. (2018); Gur-Ari et al. (2018) illuminates that training within a randomly chosen subspace facilitates parameter compression, albeit potentially at the expense of final accuracy. In subsequent work Li et al. (2022b;c), Li *et al.* successfully extracted a subspace approximating the entire parameter trajectory through principal component analysis applied to a pre-trained neural network. However, despite the richness of information captured within the aforementioned subspace, its efficacy remains primarily constrained to the early stages of pre-training. Realizing its full potential often necessitates multiple epochs.

## 3 METHODOLOGY

### 3.1 PROBLEM STATEMENT

Suppose we have $N$ clients, client $k(k \in [N])$ has its private data $\boldsymbol{x}_k$ (labelled by $y_k$) that obeys a different distribution $\mathcal{D}_k$, and the local dataset size is $n_k$. Our objective is to solve

$$\arg \min_{\mathbf{w} \in \mathbb{R}^D} \left[ F(\mathbf{w}) \triangleq \frac{1}{N} \sum_{k \in [N]} F_k(\mathbf{w}) \right], \tag{1}$$

where $F_k(\mathbf{w}) = \mathbb{E}_{(\boldsymbol{x}_k, y_k) \sim \mathcal{D}_k} [\ell_k(\mathbf{w}; (\boldsymbol{x}_k, y_k))]$ is the empirical loss of the $k$-th device, and $\mathbf{w} \in \mathbb{R}^D$ is the parameter to be optimized. We use $\mathbf{w}^*$ to represent the minimum value of the global empirical loss function.

### 3.2 CONSTRAIN THE LOCAL MODELS TO A SUBSPACE

In statistically heterogeneous scenarios, average aggregation and local training using local data cannot fully utilize the information of the global model. In addition, the huge amount of model parameters often becomes a communication bottleneck for federated learning. Our goal is to find a reasonable strategy that can not only promote parameter dimensionality reduction, but also make full use of the information from the previous global model to assist local model aggregation and training.

**Low-dimensional Subspace of Neural Network Training.** Our central concern is to reduce the communication parameters while striving to make the local model consistent with the global model. Given the redundancy in neural network representations, this characteristic can serve as the basis for mapping high-dimensional models to a low-dimensional space to find optimal solutions. In fact, previous studies have elucidated the rationale for restricting neural network training to randomly selected subspaces. Specifically, it can be expressed as:

Figure 2: We extract the top 50 principal components from the global model trajectory of ResNet-18 on Cifar100 over 200 rounds, with a cumulative percentage of 82.04%.

$$\mathbf{w}_{t+1} = \mathbf{w}_t + \mathtt{Proj}_{\mathbf{P}}(\tilde{\mathbf{g}}_t), \tag{2}$$

where $\mathbf{P} \in \mathbb{R}^{D \times R}$ is a randomly selected low-dimensional space, $D$ and $R$ represent the number of parameters and the degree of freedom of the subspace, respectively, and $\tilde{\mathbf{g}}_t$ represents the projection coefficient of the negative gradient in the low-dimensional space in the $t$-th iteration.

**Global Subspace Extraction.** We consider restricting local models to be trained in a carefully chosen low-dimensional space. Notably, this subspace is founded upon the global model, and it can approximately cover the update trajectory of the global model. Clients undertake local model training by combining global and local information, in contrast to the traditional approach where local models are exclusively trained on local data. This fusion of information mitigates the issues of local overfitting and model drift that stem from localized over-training. Consequently, the current issue that needs to be considered is how to obtain the accurate subspace that describes the global model update trajectory. This process requires two pivotal phases:

**Step 1:** Sample the first $L$ global model updates on the clients, then align the global model update into a vector $\mathbf{g}_t \in \mathbb{R}^D$, and get $\mathbf{G}_L = [\mathbf{g}_1, \mathbf{g}_2, ..., \mathbf{g}_L]$, where $\mathbf{g}_t = \mathbf{w}_{t+1} - \mathbf{w}_t$.

**Step 2:** Perform singular value decomposition on $\mathbf{G}_L$ and extract the first $R$ orthogonal bases.

Step 2 is a standard singular value decomposition problem. Its orthogonal basis can be obtained by finding the eigenvectors of $\mathbf{G}_L \mathbf{G}_L^{\mathrm{T}}$. However, for neural networks, $D$ is often very large, and it is very difficult to store and decompose $\mathbf{G}_L \mathbf{G}_L^{\mathrm{T}} \in \mathbb{R}^{D \times D}$. Since the singular value decomposition of $\mathbf{G}_L$ and $\mathbf{G}_L^{\mathrm{T}}$ are transposes of each other, we first calculate the spectral decomposition of $\mathbf{G}_L^{\mathrm{T}} \mathbf{G}_L$ to obtain the eigenvalues $\sigma_r^2$ and eigenvectors $\mathbf{v}_r$, and then calculate the orthogonal basis, $\mathbf{u}_r = \frac{1}{\sigma_r} \mathbf{G}_L \mathbf{v}_r, r = 1, ..., R$. Finally we get the global subspace $\mathbf{P} = [\mathbf{u}_1, ..., \mathbf{u}_R]$. Based on Eq. (2), the local model update can be expressed as follows:

$$\mathbf{w}_{t+1}^k - \mathbf{w}_t \approx \mathrm{Proj}_{\mathbf{P}}(\mathbf{z}_{t+1}^k - \mathbf{z}_t) = \mathrm{Proj}_{\mathbf{P}}(\tilde{\mathbf{g}}_t^k), \tag{3}$$

where $\mathbf{w}_{t+1}^k$ is the local model, and $\mathbf{z}_{t+1}^k$ is the projection coefficient of $\mathbf{w}_{t+1}^k$ on $\mathbf{P}$. The server aggregates the low-dimensional trajectory $\tilde{\mathbf{g}}_t^k$ of the client model update, $\tilde{\mathbf{g}}_t = \frac{1}{|\mathcal{M}_t|} \sum_{k \in \mathcal{M}_t} \tilde{\mathbf{g}}_t^k$.

The effectiveness of this strategy depends on whether the direction of the global optimal solution is contained within the above subspace. The absence of this critical component may result in the model not converging to satisfactory accuracy. Hence, we introduce the concept of "streaming subspace". This approach helps to track subspace changes in real-time, ensuring that the latest model updates are included.

---

**Algorithm 1** The Learning Process in FedAvg+FLSS

---

**Input:** Number of communication rounds $T$, number of parties $N$ and selected parties $M$, local steps $\tau$, learning rate $\eta$, sampling interval $s$.
**Output:** The final model $\mathbf{w}_T$.
 1: All clients to initialize their local models $\mathbf{w}_0^k$.          ▷ **Initialization Period**
 2: The clients train and communicate without FLSS for $L$ round and obtain the streaming subspace $\mathbf{P} = [\mathbf{u}_1, ..., \mathbf{u}_R]$ by $\mathrm{SLDS}_R(\cdot, \cdot)$.
 3: **for** each round $t = L, ..., T$ **do**          ▷ **Federated Learning Period**
 4:     Server samples a client subset $\mathcal{M}_t$ based on $M$.
 5:     **for** client $k \in \mathcal{M}_t$ in parallel **do**
 6:         Initialize: $\mathbf{w}_{t,1}^k \leftarrow \mathbf{w}_{t-1} + \mathrm{Proj}_{\mathbf{P}}(\tilde{\mathbf{g}}_{t-1})$ or $\mathbf{g}_{t-1}$, and update $\mathbf{P}$ by $\mathrm{SLDS}_R(\mathbf{P}, \mathbf{g}_{t-1})$.
 7:         Update the local model by batch data $\xi_{t,i}^k$: $\mathbf{w}_{t,i+1}^k \leftarrow \mathbf{w}_{t,i}^k - \eta_t \nabla F_k(\mathbf{w}_{t,i}^k, \xi_{t,i}^k)$.
 8:         Compute the local model update: $\mathbf{g}_t^k \leftarrow \mathbf{w}_{t,\tau+1}^k - \mathbf{w}_{t,1}^k$.
 9:         Transmit: $\tilde{\mathbf{g}}_t^k \leftarrow \mathrm{Proj}_{\mathbf{P}^{\mathrm{T}}}(\mathbf{g}_t^k)$ or $\mathbf{g}_t^k$ according to $\mathrm{mod}(t - L, s)$.
 10:     **end for**
 11:     Server aggregates model updates and obtains $\mathbf{g}_t$, or obtains $\tilde{\mathbf{g}}_t$ by $\tilde{\mathbf{g}}_t = \frac{1}{|\mathcal{M}_t|} \sum_{k \in \mathcal{M}_t} \tilde{\mathbf{g}}_t^k$.
 12: **end for**

---

### 3.3 STREAMING SUBSPACE

Subsequently, our focus shifts towards the acquisition of the streaming subspace. Illustrated in Fig. 3, we postulate that the subspace in the left panel approximately covers the parameter trajectories $\mathbf{w}_0, \mathbf{w}_1, ..., \mathbf{w}_3$. As the model is updated, $\mathbf{w}_4$ may deviate slightly from the subspace in the left panel.

Consequently, the current task is to identify an appropriate set of parameters to construct a subspace that can effectively encapsulate the trajectories of $\mathbf{w}_0, \mathbf{w}_1, ..., \mathbf{w}_4$.

This is accomplished by considering a sequential array, denoted as $\mathbf{G}_L, \mathbf{g}_{L+s}, \mathbf{g}_{L+2s}, ...$, of global model update. These vectors are acquired continuously and are akin to streaming data. Here, $\mathbf{G}_L$ represents the results derived from the initial $L$ samplings of the global model update. The sampling interval is set to 1 before and $s$ after the $L$-th sampling, respectively. We choose to utilize truncated singular value decomposition ($\text{SVD}_R$) for subspace extraction of global model updates, which does not require centralization and thus avoids storing previously sampled model parameters. Instead, only low-dimensional subspaces need to be preserved. Essentially, this process can be expressed as $[\hat{\mathbf{U}}, \hat{\boldsymbol{\Sigma}}, \hat{\mathbf{V}}] = \text{SVD}_R([\mathbf{G}_L, \mathbf{G}_s])$, where $\mathbf{G}_s = [\mathbf{g}_{L+s}, \mathbf{g}_{L+2s}, ...]$.

Considering that the above sequence is streaming data and the amount of model parameters is huge. The subspace needs to remain available at all times. Therefore, subspace tracking Řehůřek (2011); Eftekhari et al. (2019) is used to extract the subspace $\tilde{\mathbf{U}}$ of all model updates, $[\tilde{\mathbf{U}}, \tilde{\boldsymbol{\Sigma}}, \tilde{\mathbf{V}}] \leftarrow \text{SVD}_R([\lambda\mathbf{U}_1\boldsymbol{\Sigma}_1, \mathbf{U}_2\boldsymbol{\Sigma}_2])$, where $\lambda$ is the attenuation coefficient, assigning smaller weights to the previous subspace $\mathbf{U}_1$, and $\mathbf{G}_L = \mathbf{U}_1\boldsymbol{\Sigma}_1\mathbf{V}_1, \mathbf{G}_s = \mathbf{U}_2\boldsymbol{\Sigma}_2\mathbf{V}_2$. If $\lambda = 1$, then the subspaces $\hat{\mathbf{U}}$ and $\tilde{\mathbf{U}}$ are the same, and $\hat{\mathbf{U}} = \tilde{\mathbf{U}}\mathbf{B}$, $\mathbf{B}$ is a diagonal unitary matrix, if the non-zero singular values do not repeat, then $\mathbf{B} = \mathbf{I}_R$. Furthermore, based on the above description, we have

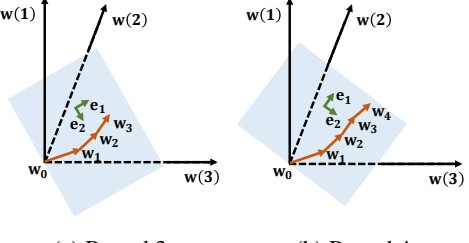

(a) Round 3        (b) Round 4

$$[\mathbf{U}, \boldsymbol{\Sigma}, \mathbf{V}] \leftarrow \text{SVD}_R([\lambda\mathbf{U}_1\boldsymbol{\Sigma}_1, \mathbf{g}_{L+s}, \mathbf{g}_{L+2s}, ...]), \tag{4}$$

Figure 3: Illustration on the update of the subspace of the global model update trajectory. The left subspace can approximately cover the trajectory of the model $\mathbf{w}_0$ to $\mathbf{w}_3$, and the right subspace can approximately cover the trajectory of the model $\mathbf{w}_0$ to $\mathbf{w}_4$.

where the subspace $\mathbf{U}$ is the same as $\hat{\mathbf{U}}$ and $\tilde{\mathbf{U}}$. For Eq. (4), if we add a global model update vector $\mathbf{g}_t$ each time and then perform $\text{SVD}_R$, it can be expressed as $\text{SVD}_R([\lambda\mathbf{U}_1\boldsymbol{\Sigma}_1, \mathbf{g}_t])$. The streaming subspace extraction process is shown in Alg. 2. The overall FLSS algorithm is shown in Alg. 1.

---

**Algorithm 2** Streaming Low Dimensional Subspace ($\text{SLDS}_R$)

---

**Input:** The degree of freedom of the subspace $R$, attenuation coefficient $\lambda$, sampling interval $s$, streaming subspace $\mathbf{P}$, global model update $\mathbf{g}_t$.
**Output:** streaming subspace $\mathbf{P}$.

1: **if** $t < L$ **then**                                                ▷ **Sampling Period**
2:      Sample global model update $\mathbf{g}_t$ to form $[\mathbf{g}_1, \mathbf{g}_2, ..., \mathbf{g}_t]$.
3: **else if** $t \geq L \,\&\, \text{mod}(t-L, s) == 0$ **then**         ▷ **Sampling and Subspace Updates**
4:      Sample global model update $\mathbf{g}_t$ to form $\mathbf{G}_t = [\lambda\mathbf{P}\boldsymbol{\Sigma}, \mathbf{g}_t]$.
5:      Perform spectral decomposition on $\mathbf{G}_t^T\mathbf{G}_t$, and obtain the largest $R$ eigenvalues $\sigma_r^2$ with the corresponding eigenvectors $\mathbf{v}_r$.
6:      Compute orthonormal basis: $\mathbf{u}_r = \frac{1}{\sigma_r}\mathbf{G}_t\mathbf{v}_r$, $\mathbf{P} = [\mathbf{u}_1, ..., \mathbf{u}_R]$, $\boldsymbol{\Sigma} = \text{diag}([\sigma_1, ...\sigma_R])$.
7: **else**
8:      $\mathbf{P} = \mathbf{P}$.

---

### 3.4 THEORETICAL ANALYSIS

Here, we consider the scenario where the clients have a global subspace $\mathbf{P}$, and local updates are aligned to this global subspace to more fully utilize the early knowledge of the global model. We then analyze the convergence of this strategy, communication and computational overhead issues.

**Assumption 3.4.1** *Loss functions $F_k$ are $L$-smooth; that is, $\forall\mathbf{v}, \mathbf{w} \in \mathbb{R}^D$, $F_k(\mathbf{v}) - F_k(\mathbf{w}) \leq \langle\mathbf{v} - \mathbf{w}, \nabla F_k(\mathbf{w})\rangle + \frac{L}{2}\|\mathbf{v} - \mathbf{w}\|_2^2, \forall k \in [N]$.*

**Assumption 3.4.2** *Loss functions $F_k$ are $\mu$-strongly convex; that is, $\forall\mathbf{v}, \mathbf{w} \in \mathbb{R}^D$, $F_k(\mathbf{v}) - F_k(\mathbf{w}) \geq \langle\mathbf{v} - \mathbf{w}, \nabla F_k(\mathbf{w})\rangle + \frac{\mu}{2}\|\mathbf{v} - \mathbf{w}\|_2^2, \forall k \in [N]$.*

**Assumption 3.4.3** *The expected squared $l_2$-norm of the stochastic gradients is bounded; that is,* $\mathbb{E}\left[\|\nabla F_k\left(\mathbf{w}_t^k, \xi_t^k\right)\|_2^2\right] \leq G^2, \forall k \in [N], \forall t.$

Assumptions 3.4.1 and 3.4.2 are standard for convergence analysis of strongly convex and smooth problems, and Assumption 3.4.3 has been made by following the works Zhang et al. (2012); Li et al. (2019); Amiri et al. (2021). We use $F^*$ and $F_k^*$ to represent the minimum value of $F(\mathbf{w})$ and $F_k(\mathbf{w})$ respectively, and use $\Gamma = F^* - \frac{1}{N}\sum_{k\in[N]} F_k^*$ to quantify the non-IID degree.

**Assumption 3.4.4** *For the global update $\mathbf{g}_t \in \mathcal{G}$, its expectation lies within the subspace; that is,* $\mathbb{E}\left[\text{Proj}(\mathbf{g}_t)\right] = \mathbb{E}\left[\mathbf{g}_t\right], \forall t$. *The detailed discussion can be found in Appendix C.5.*

**Proposition 3.4.1** *For the global update $\mathbf{g}_t \in \mathcal{G}$, the squared projection error is bounded; that is,* $\mathbb{E}\left[\|\widetilde{\text{Proj}}_\mathbf{P}(\mathbf{g}_t)\|_2^2\right] \leq \rho_R \eta_t^2 \tau^2 G^2, \rho_R = \frac{\sum_{r=R+1}^D \sigma_r^2}{\sum_{r=1}^D \sigma_r^2}, \forall t$. *The proof is in Appendix C.6.*

Assumption 3.4.4 indicates that the subspace $\mathbf{P}$ contains the expectation for global updates, where $\text{Proj}(\cdot) = \text{Proj}_{\mathbf{PP}^\mathsf{T}}(\cdot), \widetilde{\text{Proj}}_\mathbf{P}(\cdot) = \text{Proj}_{\mathbf{PP}^\mathsf{T} - \mathbf{I}_D}(\cdot)$, and $\mathcal{G}$ represents the set of global model updates. When $R = D$, the subspace $\mathbf{P}$ is full rank, which is no different from conventional training, but the storage overhead of $\mathbf{P}$ is very high. To alleviate this problem, we exploit the low-rank structure of the training trajectories to wisely choose the direction of the subspace (such as Fig. 2), thereby reducing $\rho_R$.

**Theorem 3.4.1** *Suppose that Assumptions 3.4.1 to 3.4.4 hold and a learning rate $\eta_t$ such that $0 < \eta_t \leq \min\{\frac{1}{\mu B}, \frac{1}{L(\tau+1)}\}$ is chosen, we have*

$$\mathbb{E}\left[\|\mathbf{w}_{t+1} - \mathbf{w}^*\|_2^2\right] \leq (1 - \mu\eta_t B)\mathbb{E}\left[\|\mathbf{w}_t - \mathbf{w}^*\|_2^2\right] + \eta_t^2 C, \tag{5}$$

*where*

$$B = \tau - \frac{\tau}{L(\tau+1)}, C = (2+\mu)G^2\frac{2\tau^3 + 3\tau^2 + \tau}{6} + (2L\tau^2 + 4L\tau)\Gamma + \frac{2(1+\rho_R)\tau^2 G^2}{M}. \tag{6}$$

**Corollary 3.4.1** *Suppose that Assumptions 3.4.1 to 3.4.4 hold with $\mu \geq 0$, a constant learning rate $\eta > 0$ such that $\eta \leq \frac{1}{L(\tau+1)}$, we have*

$$\mathbb{E}[F(\mathbf{w}_T)] - F^* \leq \frac{L}{2}(1-\mu\eta B)^T\mathbb{E}\left[\|\mathbf{w}_0 - \mathbf{w}^*\|_2^2\right] + \frac{L}{2}\sum_{t=1}^T \eta^2(1-\mu\eta B)^{T-t}C. \tag{7}$$

We investigate the effects of the hyperparameters $\Gamma$, $M$, and $\tau$ on the model according to Corollary 3.4.1. As indicated by the third component of $C$, reduced data heterogeneity $\Gamma$ can get closer to optimal performance. Similarly, an increase in $M$ improves performance, with $M = N$ yielding optimal results. The impact of $\tau$ on convergence is intricate, augmenting $\tau$ accelerates convergence rates as reflected by $1 - \mu\eta B$. However, this acceleration is constrained by the three terms in $C$, which ultimately limit peak performance potential.

**Corollary 3.4.2** *Let Assumptions 3.4.1 to 3.4.4 hold and $L, \mu, G, \rho_R$ be defined therein. Choose $\kappa = \frac{L}{\mu}, \gamma = \frac{2L^2(\tau+1)^2}{\mu\tau(L(\tau+1)-1)} - 1$, and the learning rate $\eta_t = \frac{2}{\mu B(\gamma+t)}$. Then* `FLSS` *satisfies*

$$\mathbb{E}[F(\mathbf{w}_T)] - F^* \leq \frac{\kappa}{\gamma + T - 1}\left(\frac{2C}{\mu B^2} + \frac{\mu(\gamma+1)}{2}\mathbb{E}\left[\|\mathbf{w}_1 - \mathbf{w}^*\|_2^2\right]\right). \tag{8}$$

It can be seen from Eq. (8) that $\lim_{T\to\infty}\mathbb{E}[F(\mathbf{w}_T)] - F^* = 0$. Notably, when $\eta_t = \frac{\beta}{t+\gamma} \leq \min\{\frac{1}{\mu B}, \frac{1}{L(\tau+1)}\}$ for some $\beta > \frac{1}{\mu B}$, `FLSS` converges to the global optimum at a rate of $\mathcal{O}(1/T)$ for strongly convex functions.

**Communication and computational overhead.** The local computation process of `FLSS` primarily encompasses subspace extraction and local model training. The computational overhead associated with subspace extraction mainly arises from the spectral decomposition of an $(R+1)\times(R+1)$ matrix, where $R$ is typically assumed to be of a small value. Consequently, the computational cost incurred by subspace extraction is negligible compared to that of local model training. In terms of communication, `FLSS` transmits a parameter vector, which has a size of either $D$ or $R$, with the communication

interval set to $s$. The average communication cost of each transmission is $\mathcal{O}((sR - R + D)/s)$. Since $R$ is much smaller than $D$, it is simplified to $\mathcal{O}(D/s)$. Additionally, FLSS mainly stores datasets, a subspace, and corresponding singular values. The sizes of the subspace and singular values are $D \times R$ and $L$ respectively, which are often negligible relative to the scale of local data.

# 4 EXPERIMENTS

## 4.1 EXPERIMENTAL SETUP

**Datasets.** In this paper, we focus on classification tasks. We conduct extensive experiments on six CV datasets FMNIST Xiao et al. (2017), Cifar10 Krizhevsky et al. (2009), Cifar100 Krizhevsky et al. (2009), Tiny-Imagenet Le & Yang (2015), and NLP dataset AG News Zhang et al. (2015). We use the ratio of 0.75 and 0.25 to divide the training data and test data. For the pathological setting, we sample disjoint data with 2/2/10/20 labels per client from 10/10/100/200 labels of FMNIST/Cifar10/Cifar100/Tiny-ImageNet. This scheme was first introduced in McMahan et al. (2017). For practical settings, we sample data from six CV datasets and AG News based on the Dirichlet distribution ($Dir(\beta)$) Kotz et al. (2019). The default $\beta$ for CV and NLP tasks are 0.1 and 1, respectively.

**Baselines.** We compare FLSS with eight federated learning baseline algorithms: FedAvg McMahan et al. (2017), FedProx Li et al. (2020b), SCAFFOLD Karimireddy et al. (2020), Moon Li et al. (2021a), FedDyn Acar et al. (2021), FedDC Gao et al. (2022), FedGen Zhu et al. (2021), FedNTD Lee et al. (2022) , and three popular communication-efficient methods: Fetchsgd Rothchild et al. (2020), Signsgd with error feedback Karimireddy et al. (2019), and STC Sattler et al. (2019).

**Hyperparameters.** For fair comparison, we set the baseline methods with local epochs as 5, the number of clients as 20 by default, the training batch size as 128, the communication rounds as 400, and the learning rate as 0.01. For the four CV datasets, we adopt the popular 4-layer CNN by default, following FedAvg, which contains two convolutional layers and two fully connected layers. In addition, we use larger models ResNet-18 He et al. (2016) and ResNet-50 He et al. (2016). For the text dataset, we used the text classification model fastText Joulin et al. (2016). All results are averages of repeated experiments with three different random seeds. All experiments are run on NVIDIA GeForce RTX 3090 GPUs.

## 4.2 PERFORMANCE COMPARISON AND ANALYSIS

### 4.2.1 PARAMETER COMPRESSION

Table 1: Average test accuracy and communication cost of different algorithms under varying degrees of heterogeneity. These include traditional and communication-efficient FL algorithms.

| Method | Cifar10 | | | | Cifar100 | | | |
|---|---|---|---|---|---|---|---|---|
| | **Com.cost** | $\beta$=0.1 | $\beta$=0.5 | $\beta$=1 | **Com.cost** | $\beta$=0.1 | $\beta$=0.5 | $\beta$=1 |
| FedAvg | 0.17 G | 57.45 | 65.29 | 67.24 | 0.18 G | 28.33 | 29.71 | 30.07 |
| FedProx | 0.17 G | 57.08 | 65.38 | 67.34 | 0.18 G | 28.42 | 29.91 | 29.94 |
| SCAFFOLD | 0.17 G | 56.30 | **67.87** | 68.01 | 0.18 G | 29.07 | 29.10 | 30.12 |
| Moon | 0.17 G | 57.65 | 65.33 | 67.41 | 0.18 G | 28.47 | 29.83 | 30.03 |
| FedDyn | 0.17 G | 55.03 | 67.58 | 67.81 | 0.18 G | 29.02 | 29.02 | 30.22 |
| FedDC | 0.17 G | 55.63 | 67.20 | 68.21 | 0.18 G | **29.14** | 29.31 | 29.98 |
| FedGen | — | 58.06 | 65.61 | 67.43 | — | 28.35 | 29.89 | 29.80 |
| FedNTD | 0.17 G | 58.07 | 65.52 | 67.53 | 0.18 G | 28.41 | 29.51 | 30.07 |
| FedAvg+FLSS | **35.14 M** | **59.44** | 67.19 | **69.82** | **37.00 M** | 28.41 | **31.47** | **31.61** |
| Fetchsgd | 43.92 M | **52.57** | 59.83 | 62.29 | 46.23 M | 22.66 | 23.98 | 25.38 |
| Signsgd+EF | 5.49 M | 51.92 | 64.27 | **66.78** | 5.78 M | 22.40 | 28.30 | 29.62 |
| STC | 5.49 M | 48.62 | 59.02 | 57.29 | 5.78 M | 21.32 | 25.30 | 24.09 |
| Sign+EF+FLSS | **1.11 M** | 52.43 | **64.69** | 63.43 | **1.16 M** | **24.11** | **29.85** | **30.61** |

We first focus on Cifar10 and Cifar100, and compare the accuracy and communication cost of different algorithms that continue to communicate for 200 rounds after sampling in Tab. 1. For FLSS,

the default settings are $L = 200$, $R = 50$, and $s = 5$. We list the accuracy of the update correction algorithms based on the same communication cost as FedAvg. After using `FLSS`, the communication cost is reduced by nearly $5\times$. Without any additional techniques (such as regularization), we can achieve lower communication cost and comparable accuracy.

In addition, to illustrate the performance of `FLSS` in the compression framework, we apply it to Signsgd+EF. In Cifar100, when we further reduce the communication cost of Signsgd+EF, the accuracy increases from $0.99\%$ to $1.71\%$.

### 4.2.2 PERFORMANCE ON VARIOUS DATASETS

Table 2: Test accuracy of different methods in two settings of statistical heterogeneity. Cifar100$^*$ and TINY$^*$ represent the test results of ResNet-18.

| Method | Pathological setting | | | | Practical setting | | | | |
|---|---|---|---|---|---|---|---|---|---|
| | FMNIST | Cifar10 | Cifar100 | Cifar100$^*$ | FMNIST | Cifar100 | TINY | TINY$^*$ | AG News |
| FedAvg | 79.02 | 55.27 | 22.98 | 23.31 | 85.06 | 28.33 | 14.26 | 14.97 | 75.13 |
| FedProx | 78.81 | 55.38 | 23.06 | 23.93 | 84.06 | 28.42 | 14.12 | 15.10 | 75.33 |
| Moon | 78.21 | 55.41 | 22.91 | 23.65 | 85.03 | 28.47 | 15.21 | 14.71 | 75.54 |
| FedDC | 79.00 | 56.76 | 23.29 | 24.24 | 83.89 | **29.14** | 15.98 | 17.75 | 79.53 |
| FedGen | 78.61 | 56.34 | 23.51 | 27.52 | 84.27 | 28.35 | 15.44 | 14.85 | 75.98 |
| FedNTD | 78.36 | 56.62 | 23.60 | 27.30 | 84.96 | 28.41 | 15.43 | 15.20 | 75.11 |
| FedAvg+`FLSS` | **81.09** | **58.89** | **25.12** | **31.46** | **86.25** | 28.41 | **17.01** | **18.56** | **81.35** |

In a comparison involving seven methods, FedAvg+`FLSS` showed good performance in both pathological and practical scenarios, as shown in Tab. 2. In the pathological setting of Cifar100, `FLSS` achieves performance improvements of $1.83\%$ compared to the baseline FedDC. Notably, applying `FLSS` to FedAvg leads to an improvement from $0.08\%$ to $6.23\%$, verifying the effectiveness of `FLSS` and that early information from the global model can assist in obtaining better solutions. In addition, it does not require further training and may be able to avoid problems such as overfitting caused by overtraining in tens of millions of dimensional parameter spaces.

### 4.3 HETEROGENEITY

We studied the performance of `FLSS` under different $\beta$ values, as shown in Tab. 1. We also compared FedAvg+`FLSS` with other methods in pathological scenarios, as shown in Tab. 2. We observe that `FLSS` outperforms most benchmarks in terms of accuracy. Notably, using the `FLSS` strategy, FedAvg improves the accuracy in the Cifar100 by $8.15\%$ and reduces the communication overhead.

### 4.4 SCALABILITY

Table 3: The test accuracy of the algorithms under different numbers of clients and different numbers of participants. We report the average running time per round for 20 clients.

| Method | $N = 20$ | | $N = 50$ | | | $N = 100$ | | | Overhead |
|---|---|---|---|---|---|---|---|---|---|
| | $M = 20$ | $M = 10$ | $M = 50$ | $M = 10$ | $M \geq 20$ | $M = 100$ | $M = 50$ | $M \geq 50$ | Avg. time |
| FedAvg | 57.45 | 56.46 | 57.10 | 56.31 | 57.65 | 50.25 | 49.70 | 49.80 | 32.26s |
| FedNTD | 58.06 | 57.10 | 57.36 | **57.25** | 57.96 | **50.93** | **50.10** | 50.12 | 80.83s |
| FedAvg+`FLSS` | **59.44** | **59.40** | **58.27** | 56.67 | **58.03** | 50.60 | 50.01 | **50.62** | 39.02s |

In real scenarios, some clients are unable to join the entire federated learning process due to client failures, network instability, etc. We simulate these scenarios by varying the number of clients $M$ in each iteration. Under different settings, `FLSS` still maintains its advantages, as shown in Tab. 3. In addition, since `FLSS` introduces operations such as subspace update and projection, we evaluate the average computational overhead of different algorithms when $M = 20$. Although `FLSS` has some shortcomings in this regard, generally, accuracy and communication efficiency are improved.

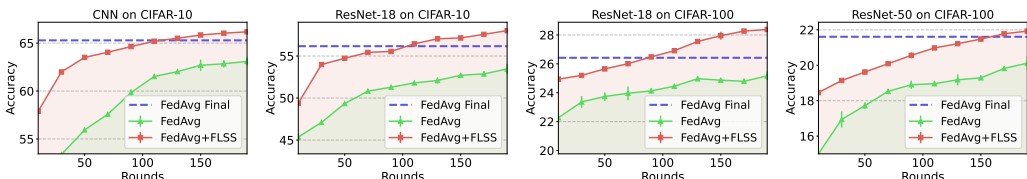

Figure 4: Performance of `FLSS` using different sampling rounds. After using the `FLSS` strategy, the accuracy of FedAvg is improved and is better than the final accuracy within 400 rounds.

## 4.5 ABLATION STUDY

**Influence of Streaming Subspace.** To verify the importance of dynamically updating subspaces, we evaluate the performance of two different strategies on Cifar100 with $\beta = 0.5$: **S**treaming **Sub**space (**S-Sub**$_{10}$, $s = 10$) and **F**ixed **Sub**space(**F-Sub**$_\infty$, $s = \infty$). We tested the performance of 200 rounds of communication, as shown in Tab. 4.

Notably, **S-Sub** shows about $1\%$ higher accuracy than **F-Sub** across three networks. This is because the fixed subspace only contains the global model information of the first 200 rounds, and the amount of information is limited. In contrast, streaming methods dynamically adapt to model changes by continuously updating the subspace, resulting in better accuracy than fixed subspace strategy.

**Effect of Subspace Degrees of Freedom.** As shown in Tab. 4, higher degrees of freedom can enhance performance in `FLSS`. Specifically, when $R = 20, 40, 50$, the test accuracy of FedAvg+`FLSS` gradually increases. This is because the higher the degree of freedom of the subspace, the more model information it contains, which is consistent with Theorem 3.4.1. However, larger $R$ also increases storage and communication requirements, so the trade-off in degrees of freedom is necessary.

Additionally, to evaluate the effect of varying the number of sampling rounds $L$, we tested the accuracy after continuing communication for 100 rounds following $L$ sampling rounds, as shown in Fig. 4. As $L$ increases, the performance of `FLSS` increases and is higher than FedAvg final accuracy.

**Effect of Attenuation Coefficient.** We studied the impact of the `FLSS` test accuracy when the attenuation coefficient $\lambda$ is 1, 0.7, and 0.5, as shown in Tab. 4. We observe that the best performance occurs when $\lambda$ is 0.7. This observation suggests that the parameter space introduced later in the streaming subspace strategy may be of greater significance for model updates. Compared with the fixed subspace strategy, it emphasizes the importance of subspace tracking. Thus, a reasonable selection of the attenuation coefficient can achieve better performance.

Table 4: Communication cost and test accuracy of FedAvg+`FLSS` under different hyperparameters, including subspace degrees of freedom, attenuation coefficient, and sampling interval.

| Model | Evaluation | $\lambda = 0.7$ | | **S-Sub**$_5$ | | | $\lambda = 1$ | | FedAvg |
|---|---|---|---|---|---|---|---|---|---|
| | | **F-Sub**$_\infty$ | **S-Sub**$_{10}$ | $\lambda = 0.5$ | $\lambda = 0.7$ | $\lambda = 1$ | $R = 20$ | $R = 40$ | |
| CNN | Com.cost (**M**) | **1e-2** | 18.50 | 37.00 | 37.00 | 37.00 | 36.98 | 36.99 | 180 |
| | Accuracy | 30.52 | 31.13 | 30.83 | **31.57** | 31.47 | 30.98 | 31.25 | 29.71 |
| ResNet-18 | Com.cost (**G**) | **1e-5** | 0.22 | 0.45 | 0.45 | 0.45 | 0.45 | 0.45 | 2.24 |
| | Accuracy | 27.98 | 28.81 | 28.91 | **29.40** | 29.32 | 28.03 | 29.10 | 26.42 |
| ResNet-50 | Com.cost (**G**) | **1e-5** | 0.47 | 0.95 | 0.95 | 0.95 | 0.95 | 0.95 | 4.74 |
| | Accuracy | 21.37 | 22.51 | 22.35 | **23.09** | 23.08 | 22.82 | 22.86 | 21.61 |

## 5 CONCLUSION

To simultaneously address the high communication costs and suboptimal performance of federated learning under statistical heterogeneity, we propose a general strategy called `FLSS`, with a theoretical guarantee. Specifically, it helps the client better utilize the early information of the global model and reduce the impact of heterogeneity on global performance by limiting local updates to the global model subspace. Notably, `FLSS` reduces communication costs by transmitting the projection coefficients of local model updates within the streaming subspace.

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

## A  RELATED WORK

### A.1  FEDERATED LEARNING

Federated learning is a distributed machine learning framework through iterative communication and computation between servers and clients. FedAvg McMahan et al. (2017) is a well-known FL method and the basic framework of many FL methods. We first introduce its main steps: (1) Server sends the current global model to clients; (2) The clients initialize the current global model as its own local model; (3) The clients train the local model on its own private data and send the trained local models to the server; (4) The server receives the client models and aggregates them to obtain the global model, and then resends it to the clients. However, the above solutions often face the problems of high communication and poor performance in heterogeneous scenarios. Therefore, a lot of work have been carried out to solve the above problems.

**Traditional Federated Learning.** Federated learning algorithms designed to enhance performance in heterogeneous environments can be divided into four different types Zhang et al. (2023a): regularization-based FL Li et al. (2020b); Acar et al. (2021); Kim et al. (2022), update correction-based FL Karimireddy et al. (2020); Gao et al. (2022); Niu & Deng (2022), model split-based FL Li et al. (2021a); Jiang et al. (2022), and knowledge distillation-based FL Zhu et al. (2021); Lee et al. (2022); Gong et al. (2022); Huang et al. (2022). In the field of regularization-based FL, FedProx Li et al. (2020b) introduces a proximal term to reduce the Euclidean distance between the global model and the local model, while FedDyn Acar et al. (2021) adopts dynamic regularization to align the local optimal point with the minimum value of the global empirical loss. For FL based on update correction, methods such as SCAFFOLD Karimireddy et al. (2020) and FedDC Gao et al. (2022) employ global gradient calibration to mitigate local model drift. However, these methods require the transmission of twice the message size required by FedAvg McMahan et al. (2017). In model split-based FL, MOON Li et al. (2021a) enhances the consistency between local and global model representations by adding a contrastive learning loss. Meanwhile, in knowledge distillation-based FL, FedGen Zhu et al. (2021) utilizes a generator trained on the server to absorb local insights and utilizes the synthesized knowledge as an inductive bias to guide the local training process. Furthermore,

FedNTD Lee et al. (2022) uses local non-true distillation to solve the problem of forgetting global information during local training.

**Communication-efficient Federated Learning.** To address the challenge of communication overhead in federated learning, many frameworks for gradient compression techniques have been proposed. Fetchsgd Rothchild et al. (2020) utilizes sketching techniques to effectively compress local gradients. Signsgd+EF Karimireddy et al. (2019) combines error feedback with 1-bit quantization, which reduces communication costs and improves the generalization ability of Signsgd. Furthermore, STC Sattler et al. (2019) is specifically designed for federated learning, combining top-k sparsity and quantization techniques to optimize data transfer. Similarly, DGC Lin et al. (2018) utilizes sparsification to preserve important gradients while minimizing bandwidth in distributed training environments. LBGM Azam et al. (2021) utilizes the low-rank characteristics of gradient space to reduce communication requirements; however, it does not fully consider the relationship between early global model information and local model updates. Although these methods have proven their feasibility in reducing communication load, their effectiveness is often limited in heterogeneous environments. This limitation is due to the stochastic nature of the compression framework and the fact that the client does not have complete information about the global model.

### A.2 TRAINING IN TINY SUBSPACE

Many studies have emphasized the inherent low-dimensional characteristics of neural networks Tuddenham et al. (2020); Vinyals & Povey (2012); Gressmann et al. (2020). A seminal study in Li et al. (2018); Gur-Ari et al. (2018) reveals that training a neural network within a randomly chosen subspace helps to achieve parameter compression, though the final accuracy may not be as high as that in the original space. The following work Gressmann et al. (2020) improved the training of fixed random subspaces by considering different layers of the network and re-drawing the random subspace at each step. Different from random subspaces, Li *et al.* Li et al. (2022b;c) successfully extracted a subspace that approximates the entire parameter trajectory by performing principal component analysis on a pre-trained neural network. Efficient dimensionality reduction is achieved by limiting the training process to this subspace.

However, although the above subspace contains model information to a certain extent, it is essentially limited to the early stage of pre-training. Often it takes multiple epochs to reach its full potential. In contrast, streaming subspace adapts to data changes by continuously updating the subspace to dynamically capture real-time model information.

## B ADDITIONAL EXPERIMENTS

### B.1 EFFECT OF PROJECTED OBJECTS

Table 5: We tested the impact of using streaming subspace on model updates or gradients respectively on algorithm performance.

| **Method** | $\text{Proj}(\mathbf{g}_t^k)$ | | $\text{Proj}(\nabla F_k \mathbf{w}_{t,i}^k)$ | | | $\text{Proj}(\mathbf{g}_t^k) + \text{Proj}(\nabla F_k(\mathbf{w}_{t,i}^k))$ | | |
|---|---|---|---|---|---|---|---|---|
| | $sr=1$ | $sr=3$ | $sr=1$ | $sr=3$ | $sr=10$ | $sr=1$ | $sr=3$ | $sr=10$ |
| FedAvg+FLSS | 57.34 | 60.02 | 55.98 | 57.58 | 56.58 | 56.10 | 57.01 | 56.28 |

We apply the streaming subspace strategy to different locations of FedAvg McMahan et al. (2017), including the model updates and gradients of the local model. Then we tested the performance of using ResNet-18 at different scaled ($sr$) learning rates in Cifar10, as shown in Tab. 5. Using the streaming subspace strategy for model updates can improve communication efficiency and performance through little additional computational overhead.

### B.2 HETEROGENEITY

To further demonstrate the performance of the FLSS-equipped algorithms on different datasets, we conduct additional experiments in heterogeneous scenarios, as shown in Tab. 6. From the results, we can see that the algorithms with FLSS outperform FedAvg McMahan et al. (2017) and Signsgd+EF

Table 6: Test accuracy on different datasets under Dirichlet distribution. Cifar100* represents using ResNet-18 on Cifar100.

| Method | FMNIST | | | Cifar100* | | | TINY | | |
|---|---|---|---|---|---|---|---|---|---|
| | $\beta$=0.1 | $\beta$=0.5 | $\beta$=1 | $\beta$=0.1 | $\beta$=0.5 | $\beta$=1 | $\beta$=0.1 | $\beta$=0.5 | $\beta$=1 |
| FedAvg | 85.06 | 91.02 | 91.18 | 23.04 | 26.42 | 27.43 | 14.26 | 15.61 | 18.47 |
| FedProx | 84.06 | 91.03 | 91.14 | 23.00 | 26.25 | 27.02 | 14.12 | 15.63 | 18.35 |
| Moon | 85.03 | 91.18 | 91.29 | 23.08 | 26.54 | 27.10 | 15.21 | 15.72 | 18.51 |
| FedDyn | 83.11 | 90.68 | 90.92 | **24.41** | 28.65 | 29.09 | 15.63 | — | **19.07** |
| FedGen | 84.27 | 91.17 | 91.25 | 23.42 | 26.24 | 27.85 | 15.44 | 15.80 | 18.72 |
| FedNTD | 84.96 | 91.15 | **91.36** | 22.84 | 26.51 | 27.15 | 15.43 | 15.77 | 18.39 |
| FedAvg+FLSS | **86.25** | **91.64** | 91.29 | 24.31 | **29.32** | **30.68** | **17.01** | **17.84** | 19.00 |
| Fetchsgd | 79.79 | 90.67 | 90.56 | 21.99 | 24.43 | 25.35 | **14.12** | 14.46 | 16.16 |
| Signsgd+EF | 80.79 | 90.76 | 90.37 | 22.57 | 25.99 | 26.01 | 14.02 | 14.20 | 16.86 |
| STC | 80.40 | 85.33 | 85.78 | 22.38 | 25.92 | **26.42** | 13.86 | 14.59 | 15.93 |
| Sign+EF+FLSS | **81.32** | **91.05** | **91.02** | **23.01** | **27.59** | 26.17 | 13.20 | **15.04** | **17.32** |

Karimireddy et al. (2019), which suggests that the FLSS strategy can effectively utilize the early knowledge of the global model to achieve better performance.

## B.3 CONVERGENCE

We present the loss throughout the training process in Fig. 5a. The experimental results confirm that the FLSS-equipped FL algorithm converges. Notably, the FLSS loss slightly increases at 200 rounds before continuing to decrease. This indicates that the model needs several rounds to adapt to the streaming subspace intervention. The loss reduction shows that constraining local model updates to subspace can continue to train and converge.

To verify the low-rank characteristics of different network update spaces, we compute the Singular Values (SV) of the other networks, as shown in Fig. 5b, Fig. 5c, and Fig. 5d. We observe that smaller networks, with larger percentages of the first few principal components, require fewer subspace degrees of freedom to approximate the update trajectory. In contrast, larger networks, such as ResNet-50, need more orthogonal bases to approximate their update trajectory.

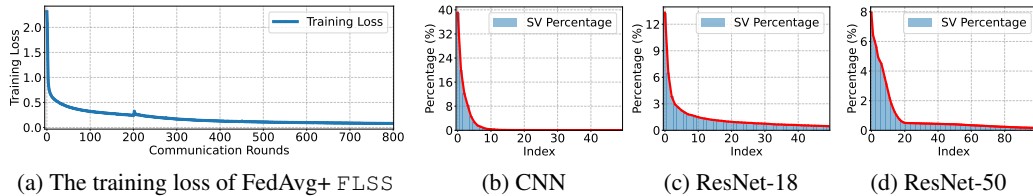

(a) The training loss of FedAvg+FLSS  (b) CNN  (c) ResNet-18  (d) ResNet-50

Figure 5: (a) is the training loss curve of FedAvg+FLSS in FMNIST. (b) to (d) are the singular value distributions of the global model update trajectory of CNN in Cifar10, ResNet-18 in Cifar10, and ResNet-50 in Cifar100, respectively.

## B.4 FEATURES VISUALIZATION

We visualize the feature representations of the different algorithms in FMNIST using t-SNE Van der Maaten & Hinton (2008) in Fig. 6. The feature representations extracted by FedAvg+FLSS become more and more distinct with iterative updates of the algorithm. Based on Fig. 6b and Fig. 6f, it can be seen that the FL algorithm equipped with FLSS ends up with more distinguishable features than those extracted by FedAvg.

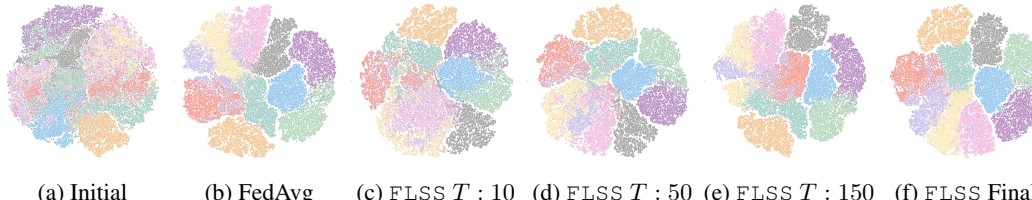

|  (a) Initial | (b) FedAvg | (c) FLSS $T:10$ | (d) FLSS $T:50$ | (e) FLSS $T:150$ | (f) FLSS Final |

Figure 6: t-SNE visualization of features extracted by the CNN model at different times on FMNIST. FLSS Final and FedAvg denote the final features with and without FLSS, respectively. $T$ denotes the number of communication rounds.

Table 7: The impact of different client local training epochs on the performance of different algorithms.

| Method | CNN | Local Epochs | | | ResNet | Local Epochs | | |
|---|---|---|---|---|---|---|---|---|
|  | Com.cost | 1 | 5 | 10 | Com.cost | 1 | 5 | 10 |
| FedAvg | 0.17 G | 59.88 | 65.29 | 65.31 | 2.23 G | 48.51 | 56.17 | 58.88 |
| FedProx | 0.17 G | 59.77 | 65.38 | 65.11 | 2.23 G | 48.44 | 56.03 | 58.27 |
| Moon | 0.17 G | 59.93 | 65.33 | 65.26 | 2.23 G | 48.74 | 56.28 | **59.24** |
| FedGen | — | 60.03 | 65.61 | 65.12 | — | 49.23 | 56.24 | 59.04 |
| FedNTD | 0.17 G | 60.42 | 65.52 | 65.15 | 2.23 G | 49.64 | 56.42 | 59.06 |
| FedAvg+FLSS | **35.14 M** | **62.31** | **67.19** | **65.53** | **0.45 G** | **51.92** | **57.34** | 58.93 |
| Fetchsgd | 43.92 M | 52.20 | 59.83 | 58.57 | 0.56 G | — | 54.18 | 56.04 |
| Signsgd+EF | 5.49 M | 59.21 | 64.27 | 63.95 | 69.88 M | 47.22 | 54.23 | 55.68 |
| STC | 5.49 M | **59.55** | 59.02 | 62.38 | 69.88 M | 47.91 | **55.03** | 55.35 |
| Sign+EF+FLSS | **1.11 M** | 59.01 | **64.69** | **64.06** | **14.0 M** | **49.02** | 54.77 | **56.81** |

## B.5 DIFFERENT LOCAL EPOCHS

Increasing local epochs results in higher computational costs but reduces the number of communication rounds. We evaluate the performance of CNN and ResNet-18 over 400 rounds on Cifar10 with $\beta = 0.5$, as shown in Tab. 7. Across different local epochs settings, FLSS performs better than most baselines. Notably, FLSS shows more significant performance improvement with fewer local epochs. Specifically, with 1 local epoch, FedAvg combined with FLSS achieves improvements of 2.43% and 3.41%, respectively.

## B.6 COMPARISON WITH OTHER BASELINES

Table 8: Average test accuracy and communication cost of different algorithms under varying degrees of heterogeneity.

| Method | Cifar10 | | | Cifar100 | | |
|---|---|---|---|---|---|---|
|  | $\beta=0.1$ | $\beta=0.5$ | $\beta=1$ | $\beta=0.1$ | $\beta=0.5$ | $\beta=1$ |
| LBGM | 56.37(0.21G) | 64.42(0.20G) | 65.15(0.20G) | 27.70(0.27G) | 28.81(0.27G) | 30.01(0.26 G) |
| FedAvg+FLSS | 59.44(0.21G) | 67.19(0.21G) | 69.82(0.21G) | 28.41(0.22G) | 31.47(0.22G) | 31.61(0.22G) |

We compared the performance and total communication cost of 400 rounds between LBGM and FLSS, as shown in Tab. 8. LBGM is a low-rank method based on gradient space, focusing on local training trajectories, while FLSS targets low-rank properties of the global model. FLSS projects local updates onto the global subspace to filter out harmful components. Additionally, LBGM emphasizes a single gradient direction early in training, while FLSS uses all early training information, unifying them into a low-rank subspace.

## C CONVERGENCE OF FLSS

### C.1 NOTATION

We defined the local model update at device $k$ as $\mathbf{g}_t^k = \mathbf{w}_{t+1}^k - \mathbf{w}_t$, We define the low-dimensional trajectory of the local model updated on device $k$ within subspace $\mathbf{P}$ as $\hat{\mathbf{g}}_t^k$, $\hat{\mathbf{g}}_t^k = \mathtt{Proj}_{\mathbf{P}}(\mathtt{Proj}_{\mathbf{P}^\top}(\mathbf{w}_{t+1}^k - \mathbf{w}_t))$. The global model parameters updated as $\mathbf{w}_{t+1} = \mathbf{w}_t + \frac{1}{M}\sum_{k \in \mathcal{M}_t}\hat{\mathbf{g}}_t^k$, we define the following auxiliary variable: $\mathbf{v}_{t+1} = \mathbf{w}_t + \frac{1}{N}\sum_{k=1}^N \hat{\mathbf{g}}_t^k$. We have

$$
\begin{aligned}
\|\mathbf{w}_{t+1} - \mathbf{w}^*\|_2^2 &= \|\mathbf{w}_{t+1} - \mathbf{v}_{t+1} + \mathbf{v}_{t+1} - \mathbf{w}^*\|_2^2 \\
&= \|\mathbf{w}_{t+1} - \mathbf{v}_{t+1}\|_2^2 + \|\mathbf{v}_{t+1} - \mathbf{w}^*\|_2^2 + 2\langle\mathbf{w}_{t+1} - \mathbf{v}_{t+1}, \mathbf{v}_{t+1} - \mathbf{w}^*\rangle.
\end{aligned}
\tag{9}
$$

In the following, we bound the average of the terms on the right hand side (RHS).

### C.2 KEY LEMMAS

**Lemma C.2.1** *Suppose that Assumptions 3.4.1 to 3.4.4 hold, the difference between $\mathbf{w}_{t+1}$ and $\mathbf{v}_{t+1}$ can be bounded*

$$
\mathbb{E}\left[\|\mathbf{w}_{t+1} - \mathbf{v}_{t+1}\|_2^2\right] \le \frac{2\eta_t^2\tau^2 G^2}{M} + \frac{2\sum_{r=R+1}^D \sigma_r^2}{M\sum_{r=1}^D \sigma_r^2}\eta_t^2\tau^2 G^2.
$$

*Proof. See Appendix C.7.*

**Lemma C.2.2** *Suppose that Assumptions 3.4.1 to 3.4.4 hold, the upper bound of $\mathbb{E}\left[\|\mathbf{v}_{t+1} - \mathbf{w}^*\|_2^2\right]$ is as follows*

$$
\begin{aligned}
\mathbb{E}\left[\|\mathbf{v}_{t+1} - \mathbf{w}^*\|_2^2\right] &\le (1 - \mu\eta_t\tau(1-\eta_t))\mathbb{E}\left[\|\mathbf{w}_t - \mathbf{w}^*\|_2^2\right] \\
&\quad + (2+\mu)\eta_t^2 G^2\frac{\tau(\tau+1)(2\tau+1)}{6} + (2L\eta_t^2\tau^2 + 4L\eta_t^2\tau)\Gamma.
\end{aligned}
$$

*Proof. See Appendix C.8.*

**Lemma C.2.3** *Let $\mathbb{E}_{\mathcal{M}_t}$ denote expectation over the device scheduling randomness at the global iteration $t$. We have $\mathbb{E}_{\mathcal{M}_t}[\mathbf{w}_{t+1}] = \mathbf{v}_{t+1}$, from which it follows that*

$$
\mathbb{E}_{\mathcal{M}_t}[\langle\mathbf{w}_{t+1} - \mathbf{v}_{t+1}, \mathbf{v}_{t+1} - \mathbf{w}^*\rangle] = 0.
$$

*Proof. Due to the randomness of the device scheduling policy and the scheduling update of each device appears $\binom{N-1}{M-1}$ times, it follows that*

$$
\mathbb{E}_{\mathcal{M}_t}\left[\frac{1}{M}\sum_{k \in \mathcal{M}_t}\hat{\mathbf{g}}_t^k\right] = \frac{\binom{N-1}{M-1}}{M\binom{N}{M}}\sum_{k=1}^N \hat{\mathbf{g}}_t^k = \frac{1}{N}\sum_{k=1}^N \hat{\mathbf{g}}_t^k.
\tag{10}
$$

### C.3 THEOREMS

**Theorem C.3.1** *Suppose that Assumptions 3.4.1 to 3.4.4 hold and a learning rate $\eta_t$ such that $0 < \eta_t \le \min\{\frac{1}{\mu B}, \frac{1}{L(\tau+1)}\}$ is chosen, we have*

$$
\mathbb{E}\left[\|\mathbf{w}_{t+1} - \mathbf{w}^*\|_2^2\right] \le (1 - \mu\eta_t B)\mathbb{E}\left[\|\mathbf{w}_t - \mathbf{w}^*\|_2^2\right] + \eta_t^2 C,
\tag{11}
$$

*where*

$$
B = \tau - \frac{\tau}{L(\tau+1)}, C = (2+\mu)G^2\frac{2\tau^3 + 3\tau^2 + \tau}{6} + (2L\tau^2 + 4L\tau)\Gamma + \frac{2(1+\rho_R)\tau^2 G^2}{M}.
\tag{12}
$$

*Proof. See Appendix C.9.*

## C.4 COROLLARIES

**Corollary C.4.1** *Suppose that Assumptions 3.4.1 to 3.4.4 hold with $\mu \geq 0$, a constant learning rate $\eta > 0$ such that $\eta \leq \frac{1}{L(\tau+1)}$, we have*

$$\mathbb{E}[F(\mathbf{w}_T)] - F^* \leq \frac{L}{2}(1 - \mu\eta B)^T \mathbb{E}\left[\|\mathbf{w}_0 - \mathbf{w}^*\|_2^2\right] + \frac{L}{2}\sum_{t=1}^{T} \eta^2 (1 - \mu\eta B)^{T-t} C. \tag{13}$$

*Proof. See Appendix C.10.1.*

**Corollary C.4.2** *Let Assumptions 3.4.1 to 3.4.4 hold and $L, \mu, G, \rho_R$ be defined therein. Choose $\kappa = \frac{L}{\mu}$, $\gamma = \frac{2L^2(\tau+1)^2}{\mu\tau(L(\tau+1)-1)} - 1$, and the learning rate $\eta_t = \frac{2}{\mu B(\gamma+t)}$. Then FLSS satisfies*

$$\mathbb{E}[F(\mathbf{w}_T)] - F^* \leq \frac{\kappa}{\gamma + T - 1}\left(\frac{2C}{\mu B^2} + \frac{\mu(\gamma+1)}{2}\mathbb{E}\left[\|\mathbf{w}_1 - \mathbf{w}^*\|_2^2\right]\right). \tag{14}$$

*Proof. See Appendix C.10.2.*

## C.5 DISCUSSION ON ASSUMPTION 3.4.4

We define $\mathcal{G}$ to be the set consisting of global model updates and $\mathbf{G} \in \mathbb{R}^{D \times J}$ to be the matrix consisting of the set of global model updates $\mathcal{G}$. In the experiments, $\mathbf{G} \in \mathbb{R}^{D \times J}$ can be obtained by sampling the global updates. A truncated singular value decomposition of $\mathbf{G}$ of rank $R$ yields $\mathbf{P}$, whose singular values are $\sigma_1, ..., \sigma_D$. Based on the linearity property of expectation, we have

$$\mathbb{E}\left[\text{Proj}(\mathbf{g}_t)\right] = \mathbb{E}[\mathbf{P}\mathbf{P}^{\text{T}}\mathbf{g}_t] = \mathbf{P}\mathbf{P}^{\text{T}}\mathbb{E}[\mathbf{g}_t] = \text{Proj}(\mathbb{E}[\mathbf{g}_t]). \tag{15}$$

Due to the low-rank character of the global model update space, the last few singular values are small and the corresponding dimensions are almost null space. In Eq. (15), it is assumed that the expectation of the global model update $\mathbb{E}[\mathbf{g}_t]$ will be contained within the subspace $\mathbf{P}$, so $\mathbb{E}[\text{Proj}(\mathbf{g}_t)] = \mathbb{E}[\mathbf{g}_t]$.

## C.6 PROOFS OF PROPOSITION 3.4.1

For the global update $\mathbf{g}_t \in \mathcal{G}$, we compute the expectation of the squared projection error:

$$\mathbb{E}_{\mathbf{g}_t \in \mathcal{G}}(\|\mathbf{g}_t - \text{Proj}(\mathbf{g}_t)\|^2) = \frac{1}{J}\sum_{j=1}^{J}\|\mathbf{G}_j - \mathbf{P}\mathbf{P}^{\text{T}}\mathbf{G}_j\|^2. \tag{16}$$

Using trace properties, we have

$$\frac{1}{J}\sum_{i=1}^{J}\|\mathbf{G}_j - \mathbf{P}\mathbf{P}^{\text{T}}\mathbf{G}_j\|^2 = \frac{1}{J}\text{tr}(\mathbf{G}^{\text{T}}(\mathbf{I}_D - \mathbf{P}\mathbf{P}^{\text{T}})\mathbf{G}) = \frac{1}{J}\text{tr}((\mathbf{I}_D - \mathbf{P}\mathbf{P}^{\text{T}})\mathbf{G}\mathbf{G}^{\text{T}}). \tag{17}$$

Since $\mathbf{I}_D - \mathbf{P}\mathbf{P}^{\text{T}}$ projected $\mathbf{G}\mathbf{G}^{\text{T}}$ to a space orthogonal to the columns of $\mathbf{P}$, we have

$$\mathbb{E}(\|\mathbf{g}_t - \text{Proj}(\mathbf{g}_t)\|^2) = \frac{1}{J}\sum_{r=R+1}^{D}\sigma_r^2 \leq \frac{\sum_{r=R+1}^{D}\sigma_r^2}{\sum_{r=1}^{D}\sigma_r^2}\eta_t^2\tau^2 G^2. \tag{18}$$

The last inequality is due to Assumption 3.4.3, $\mathbb{E}\left[\|\nabla F_k\left(\mathbf{w}_t^k, \xi_t^k\right)\|_2^2\right] \leq G^2$, so that $\mathbb{E}\left[\|\mathbf{g}_t\|_2^2\right] \leq \eta_t^2\tau^2 G^2$ and $\|\mathbf{G}\|_2^2 \leq J\eta_t^2\tau^2 G^2$.

## C.7 PROOF OF LEMMA C.2.1

According to the definitions, $\mathbf{w}_{t+1} = \mathbf{w}_t + \frac{1}{M}\sum_{k\in\mathcal{M}_t}\hat{\mathbf{g}}_t^k$, $\mathbf{v}_{t+1} = \mathbf{w}_t + \frac{1}{N}\sum_{k=1}^N\hat{\mathbf{g}}_t^k$, $i_m \in \mathcal{M}_t$, and $\hat{\mathbf{g}}_t \triangleq \frac{1}{N}\sum_{k=1}^N\hat{\mathbf{g}}_t^k$. Taking the expectation of the first term of Eq. (9), we have

$$
\mathbb{E}_{\mathcal{M}_t}\left[\|\mathbf{w}_{t+1} - \mathbf{v}_{t+1}\|_2^2\right] = \mathbb{E}_{\mathcal{M}_t}\left[\left\|\frac{1}{M}\sum_{m=1}^M\left(\hat{\mathbf{g}}_t^{i_m} - \hat{\mathbf{g}}_t\right)\right\|_2^2\right]
$$

$$
= \frac{1}{M^2}\mathbb{E}_{\mathcal{M}_t}\left[\sum_{m=1}^M\left\|\hat{\mathbf{g}}_t^{i_m} - \hat{\mathbf{g}}_t\right\|_2^2 + \sum_{m=1}^M\sum_{m'=1,m'\neq m}^M\langle\hat{\mathbf{g}}_t^{i_m} - \hat{\mathbf{g}}_t, \hat{\boldsymbol{g}}_t^{i_{m'}} - \hat{\mathbf{g}}_t\rangle\right]. \tag{19}
$$

Due to the symmetry, it follows that

$$
\mathbb{E}_{\mathcal{M}_t}\left[\sum_{m=1}^M\left\|\hat{\mathbf{g}}_t^{i_m} - \hat{\mathbf{g}}_t\right\|_2^2\right] = \frac{\binom{N-1}{M-1}}{\binom{N}{M}}\sum_{k=1}^N\|\hat{\mathbf{g}}_t^k - \hat{\mathbf{g}}_t\|_2^2 = \frac{M}{N}\sum_{k=1}^N\left\|\hat{\mathbf{g}}_t^k - \hat{\mathbf{g}}_t\right\|_2^2, \tag{20}
$$

where the first equality is because there are $\binom{N}{M}$ choices in selecting $M$ from $N$ clients. For each index $k$, $k \in [N]$, the number of times is selected is $\binom{N-1}{M-1}$.

$$
\mathbb{E}_{\mathcal{M}_t}\left[\sum_{m=1}^M\sum_{m'=1,m'\neq m}^M\langle\hat{\mathbf{g}}_t^{i_m} - \hat{\mathbf{g}}_t, \hat{\boldsymbol{g}}_t^{i_{m'}} - \hat{\mathbf{g}}_t\rangle\right] = \frac{\binom{N-2}{M-2}}{\binom{N}{M}}\sum_{k=1}^K\sum_{\substack{k'=1\\k'\neq k}}^N\langle\hat{\mathbf{g}}_t^k - \hat{\mathbf{g}}_t, \hat{\boldsymbol{g}}_t^{k'} - \hat{\mathbf{g}}_t\rangle
$$

$$
= -\frac{\binom{N-2}{M-2}}{\binom{N}{M}}\sum_{k=1}^N\left\|\hat{\mathbf{g}}_t^k - \hat{\mathbf{g}}_t\right\|_2^2 \leq 0. \tag{21}
$$

where the first equality is because, for each particular index pair $(k,k')$, $k' \in [N]$, $k \neq k'$, the number of times is selected is $\binom{N-2}{M-2}$, and the second equality is because $\left\|\sum_{k=1}^N\left(\hat{\mathbf{g}}_t^k - \hat{\mathbf{g}}_t\right)\right\|_2^2 = 0$. Substituting Eq. (20) and Eq. (21) into Eq. (19) yields

$$
\mathbb{E}\left[\|\mathbf{w}_{t+1} - \mathbf{v}_{t+1}\|_2^2\right]
$$

$$
= \frac{1}{NM}\sum_{k=1}^N\mathbb{E}\left[\|\hat{\mathbf{g}}_t^k - \hat{\mathbf{g}}_t\|_2^2\right] + \frac{\binom{N-2}{M-2}}{M^2\binom{N}{M}}\sum_{k=1}^N\sum_{\substack{k'=1\\k'\neq k}}^N\langle\hat{\mathbf{g}}_t^k - \hat{\mathbf{g}}_t, \hat{\boldsymbol{g}}_t^{k'} - \hat{\mathbf{g}}_t\rangle \leq \frac{1}{NM}\sum_{k=1}^N\mathbb{E}\left[\|\hat{\mathbf{g}}_t^k - \hat{\mathbf{g}}_t\|_2^2\right]
$$

$$
= \frac{1}{NM}\left(\sum_{k=1}^N\mathbb{E}\left[\|\hat{\mathbf{g}}_t^k\|_2^2\right] - \mathbb{E}\left[\|\hat{\mathbf{g}}_t\|_2^2\right]\right) \leq \frac{1}{NM}\sum_{k=1}^N\mathbb{E}\left[\|\hat{\mathbf{g}}_t^k\|_2^2\right] \leq \frac{1}{NM}\sum_{k=1}^N\mathbb{E}\left[\|\mathbf{g}_t^k + \mathbf{e}_t^k\|_2^2\right]
$$

$$
\leq \frac{2\eta_t^2\tau^2 G^2}{M} + \frac{2\sum_{r=R+1}^D\sigma_r^2}{M\sum_{r=1}^D\sigma_r^2}\eta_t^2\tau^2 G^2. \tag{22}
$$

## C.8 PROOF OF LEMMA C.2.2

According to the definition of $\mathbf{v}_{t+1}$, $\mathbf{v}_{t+1} = \mathbf{w}_t + \frac{1}{N}\sum_{k=1}^N\hat{\mathbf{g}}_t^k$, taking the expectation and expanding the second term of the Eq. (9), we have

$$
\mathbb{E}\left[\|\mathbf{v}_{t+1} - \mathbf{w}^*\|_2^2\right] = \underbrace{\mathbb{E}\left[\|\mathbf{w}_t - \mathbf{w}^*\|_2^2\right]}_{A_1} + \underbrace{\mathbb{E}\left[\left\|\frac{1}{N}\sum_{k=1}^N\hat{\mathbf{g}}_t^k\right\|_2^2\right]}_{A_2} + \underbrace{2\mathbb{E}\left[\left\langle\mathbf{w}_t - \mathbf{w}^*, \frac{1}{N}\sum_{k=1}^N\hat{\mathbf{g}}_t^k\right\rangle\right]}_{A_3}. \tag{23}
$$

For $A_2$, due to the convexity of $\| \cdot \|_2^2$ and the $L$-smoothness of $F_k(\cdot)$, $\left\| \nabla F_k \left( \mathbf{w}_{t,i}^k, \xi_{t,i}^k \right) \right\|^2 \leq 2L \left( F_k(\mathbf{w}_{t,i}^k) - F_k^* \right)$, we have

$$
A_2 \leq \frac{1}{N} \sum_{k=1}^N \mathbb{E} \left[ \| \hat{\mathbf{g}}_t^k \|_2^2 \right] \leq \frac{1}{N} \sum_{k=1}^N \mathbb{E} \left[ \| \mathbf{g}_t^k \|_2^2 \right] = \frac{\eta_t^2}{N} \sum_{k=1}^N \mathbb{E} \left[ \left\| \sum_{i=1}^\tau \nabla F_k \left( \mathbf{w}_{t,i}^k, \xi_{t,i}^k \right) \right\|_2^2 \right]
$$

$$
\leq \frac{\eta_t^2 \tau}{N} \sum_{k=1}^N \sum_{i=1}^\tau \mathbb{E} \left[ \left\| \nabla F_k \left( \mathbf{w}_{t,i}^k, \xi_{t,i}^k \right) \right\|_2^2 \right] \leq \frac{2L \eta_t^2 \tau}{N} \sum_{k=1}^N \sum_{i=1}^\tau \mathbb{E} \left[ F_k(\mathbf{w}_{t,i}^k) - F_k^* \right]. \tag{24}
$$

For $A_3$, according to Assumption 3.4.4 and the definition of $\hat{\mathbf{g}}_t^k$, we can know that $\hat{\mathbf{g}}_t^k = \mathrm{Proj}_{\mathbf{P}}(\mathrm{Proj}_{\mathbf{P}^\top}(\mathbf{g}_t^k))$, and $\mathrm{Proj}_{\mathbf{P}}(\mathrm{Proj}_{\mathbf{P}^\top}(\mathbf{w}_t - \mathbf{w}^*) = \mathbf{w}_t - \mathbf{w}^* + \epsilon_t$. We have

$$
2\mathbb{E} \left[ \langle \mathbf{w}_t - \mathbf{w}^*, \frac{1}{N} \sum_{k=1}^N \hat{\mathbf{g}}_t^k \rangle \right] = \underbrace{\frac{2\eta_t}{N} \sum_{k=1}^N \mathbb{E} \left[ \langle \mathbf{w}^* - \mathbf{w}_t, \sum_{i=1}^\tau \nabla F_k \left( \mathbf{w}_{t,i}^k, \xi_{t,i}^k \right) \rangle \right]}_{B_1}. \tag{25}
$$

For $B_1$, we split $\mathbf{w}^* - \mathbf{w}_t$ into $\mathbf{w}^* - \mathbf{w}_{t,i}^k$ and $\mathbf{w}_{t,i}^k - \mathbf{w}_t$, so $B_1$ can be split into two items: $C_1 = \frac{2\eta_t}{K} \sum_{k=1}^N \sum_{i=1}^\tau \mathbb{E} \left[ \langle \mathbf{w}_{t,i}^k - \mathbf{w}_t, \nabla F_k \left( \mathbf{w}_{t,i}^k, \xi_{t,i}^k \right) \rangle \right]$, $C_2 = \frac{2\eta_t}{K} \sum_{k=1}^N \sum_{i=1}^\tau \mathbb{E} \left[ \langle \mathbf{w}^* - \mathbf{w}_{t,i}^k, \nabla F_k \left( \mathbf{w}_{t,i}^k, \xi_{t,i}^k \right) \rangle \right]$. So next we calculate the upper bounds of these two terms respectively. To bound $C_1$, we have

$$
C_1 \leq \frac{\eta_t}{N} \sum_{k=1}^N \sum_{i=1}^\tau \mathbb{E} \left[ \frac{1}{\eta_t} \left\| \mathbf{w}_{t,i}^k - \mathbf{w}_t \right\|_2^2 + \eta_t \left\| \nabla F_k \left( \mathbf{w}_{t,i}^k, \xi_{t,i}^k \right) \right\|_2^2 \right]
$$

$$
\leq \frac{1}{N} \sum_{k=1}^N \sum_{i=1}^\tau \mathbb{E} \left[ \left\| \mathbf{w}_{t,i}^k - \mathbf{w}_t \right\|_2^2 \right] + \frac{2L \eta_t^2}{N} \sum_{k=1}^N \sum_{i=1}^\tau \mathbb{E} \left[ F_k(\mathbf{w}_{t,i}^k) - F_k^* \right], \tag{26}
$$

where the first inequality is by Cauchy-Schwarz inequality, and the second inequality is by the $L$-smoothness of $F_k(\cdot)$, $\left\| \nabla F_k \left( \mathbf{w}_{t,i}^k, \xi_{t,i}^k \right) \right\|^2 \leq 2L \left( F_k(\mathbf{w}_{t,i}^k) - F_k^* \right)$. To bound $C_2$, we have

$$
C_2 = \frac{2\eta_t}{N} \sum_{k=1}^N \sum_{i=1}^\tau \mathbb{E} \left[ \langle \mathbf{w}^* - \mathbf{w}_{t,i}^k, \nabla F_k \left( \mathbf{w}_{t,i}^k \right) \rangle \right]
$$

$$
\leq \frac{2\eta_t}{N} \sum_{k=1}^N \sum_{i=1}^\tau \mathbb{E} \left[ F_k(\mathbf{w}^*) - F_k(\mathbf{w}_{t,i}^k) - \frac{\mu}{2} \left\| \mathbf{w}_{t,i}^k - \mathbf{w}^* \right\|_2^2 \right], \tag{27}
$$

where the first equality is by $\mathbb{E}_\xi \left[ \nabla F_k \left( \mathbf{w}_t, \xi_{t,i}^k \right) \right] = \nabla F_k \left( \mathbf{w}_t \right), \forall i, k, t$ and the first inequality is by the fact that $F_k$ is $\mu$-strongly convex.

For $A_3$, substituting Eq. (26) and Eq. (27) into Eq. (25), we have

$$
2\mathbb{E} \left[ \langle \mathbf{w}_t - \mathbf{w}^*, \frac{1}{N} \sum_{k=1}^N \hat{\mathbf{g}}_t^k \rangle \right] \leq \frac{1}{N} \sum_{k=1}^N \sum_{i=1}^\tau \mathbb{E} \left[ \left\| \mathbf{w}_{t,i}^k - \mathbf{w}_t \right\|_2^2 \right] + \frac{2L \eta_t^2}{N} \sum_{k=1}^N \sum_{i=1}^\tau \mathbb{E} \left[ F_k(\mathbf{w}_{t,i}^k) - F_k^* \right]
$$

$$
+ \frac{2\eta_t}{N} \sum_{k=1}^N \sum_{i=1}^\tau \mathbb{E} \left( F_k(\mathbf{w}^*) - F_k(\mathbf{w}_{t,i}^k) \right) - \frac{\mu \eta_t}{N} \sum_{k=1}^N \sum_{i=1}^\tau \mathbb{E} \left[ \left\| \mathbf{w}_{t,i}^k - \mathbf{w}^* \right\|_2^2 \right]. \tag{28}
$$

For $\mathbb{E}\left[\|\mathbf{v}_{t+1} - \mathbf{w}^*\|_2^2\right]$, substituting Eq. (28) and Eq. (24) into Eq. (23), we have

$$
\mathbb{E}\left[\|\mathbf{v}_{t+1} - \mathbf{w}^*\|_2^2\right]
$$

$$
\leq \mathbb{E}\left[\|\mathbf{w}_t - \mathbf{w}^*\|_2^2\right] + \frac{2L\eta_t^2\tau}{N}\sum_{k=1}^{N}\sum_{i=1}^{\tau}\mathbb{E}\left[F_k(\mathbf{w}_{t,i}^k) - F_k^*\right] + \frac{1}{N}\sum_{k=1}^{N}\sum_{i=1}^{\tau}\mathbb{E}\left[\|\mathbf{w}_{t,i}^k - \mathbf{w}_t\|_2^2\right]
$$

$$
+ \frac{2L\eta_t^2}{N}\sum_{k=1}^{N}\sum_{i=1}^{\tau}\mathbb{E}\left[F_k(\mathbf{w}_{t,i}^k) - F_k^*\right] + \frac{2\eta_t}{N}\sum_{k=1}^{N}\sum_{i=1}^{\tau}\mathbb{E}\left(F_k(\mathbf{w}^*) - F_k(\mathbf{w}_{t,i}^k)\right)
$$

$$
- \frac{\mu\eta_t}{N}\sum_{k=1}^{N}\sum_{i=1}^{\tau}\mathbb{E}\left[\|\mathbf{w}_{t,i}^k - \mathbf{w}^*\|_2^2\right] \tag{29}
$$

$$
= \mathbb{E}\left[\|\mathbf{w}_t - \mathbf{w}^*\|_2^2\right] \underbrace{- \frac{\mu\eta_t}{N}\sum_{k=1}^{N}\sum_{i=1}^{\tau}\mathbb{E}\left[\|\mathbf{w}_{t,i}^k - \mathbf{w}^*\|_2^2\right] + \frac{1}{N}\sum_{k=1}^{N}\sum_{i=1}^{\tau}\mathbb{E}\left[\|\mathbf{w}_{t,i}^k - \mathbf{w}_t\|_2^2\right]}_{D_1}
$$

$$
\underbrace{+ \frac{2L\eta_t^2(\tau+1)}{N}\sum_{k=1}^{N}\sum_{i=1}^{\tau}\mathbb{E}\left[F_k(\mathbf{w}_{t,i}^k) - F_k^*\right] - \frac{2\eta_t}{N}\sum_{k=1}^{N}\sum_{i=1}^{\tau}\mathbb{E}\left(F_k(\mathbf{w}_{t,i}^k - F_k(\mathbf{w}^*))\right)}_{D_2}.
$$

To bound $D_1$, we first calculate the upper bound of $-\|\mathbf{w}_{t,i}^k - \mathbf{w}^*\|_2^2$

$$
-\|\mathbf{w}_{t,i}^k - \mathbf{w}^*\|_2^2 = -\|\mathbf{w}_{t,i}^k - \mathbf{w}_t\|_2^2 - \|\mathbf{w}_t - \mathbf{w}^*\|_2^2 - 2\langle\mathbf{w}_{t,i}^k - \mathbf{w}_t, \mathbf{w}_t - \mathbf{w}^*\rangle
$$

$$
\leq -\|\mathbf{w}_{t,i}^k - \mathbf{w}_t\|_2^2 - \|\mathbf{w}_t - \mathbf{w}^*\|_2^2 + \frac{1}{\eta_t}\|\mathbf{w}_{t,i}^k - \mathbf{w}_t\|_2^2 + \eta_t\|\mathbf{w}_t - \mathbf{w}^*\|_2^2
$$

$$
= -(1-\eta_t)\|\mathbf{w}_t - \mathbf{w}^*\|_2^2 + \left(\frac{1}{\eta_t} - 1\right)\|\mathbf{w}_{t,i}^k - \mathbf{w}_t\|_2^2, \tag{30}
$$

where the first inequality is by Cauchy-Schwarz inequality. We next aim to bound $D_2$. We define $\gamma_t = 2\eta_t(1 - L\eta_t\tau - L\eta_t)$. Let $\gamma_t \geq 0$, we have $\eta_t \leq \frac{1}{L(\tau+1)}, \gamma_t \leq 2\eta_t$. We define $\Gamma = F^* - \frac{1}{N}\sum_{k=1}^{N}F_k^*$, which is a measure of non-IID degree. Then we have

$$
D_2 = \frac{2L\eta_t^2(\tau+1)}{N}\sum_{k=1}^{N}\sum_{i=1}^{\tau}\mathbb{E}\left[F_k(\mathbf{w}_{t,i}^k) - F_k^*\right] - \frac{2\eta_t}{N}\sum_{k=1}^{N}\sum_{i=1}^{\tau}\mathbb{E}\left(F_k(\mathbf{w}_{t,i}^k - F_k(\mathbf{w}^*))\right)
$$

$$
= \underbrace{-\frac{\gamma_t}{N}\sum_{k=1}^{N}\sum_{i=1}^{\tau}\mathbb{E}\left[F_k(\mathbf{w}_{t,i}^k) - F^*\right]}_{E} + \frac{2L\eta_t^2(\tau+1)}{N}\sum_{k=1}^{N}\sum_{i=1}^{\tau}\mathbb{E}\left[F^* - F_k^*\right]. \tag{31}
$$

To bound $E$, considering $\gamma_t \geq 0$, we need to obtain the lower bound of $\frac{1}{N}\sum_{k=1}^{N}\sum_{i=1}^{\tau}\mathbb{E}\left[F_k(\mathbf{w}_{t,i}^k) - F^*\right]$. Then, we split $F_k(\mathbf{w}_{t,i}^k) - F^*$ into $F_k(\mathbf{w}_{t,i}^k) - F_k(\mathbf{w}_t)$ and $F_k(\mathbf{w}_t) - F^*$, and take the expectations of them, respectively. We first calculate the lower bound of $\frac{1}{N}\sum_{k=1}^{N}\sum_{i=1}^{\tau}\mathbb{E}\left[F_k(\mathbf{w}_{t,i}^k) - F_k(\mathbf{w}_t)\right]$:

$$
\frac{1}{N}\sum_{k=1}^{N}\sum_{i=1}^{\tau}\mathbb{E}\left[F_k(\mathbf{w}_{t,i}^k) - F_k(\mathbf{w}_t)\right] \geq \frac{1}{N}\sum_{k=1}^{N}\sum_{i=1}^{\tau}\mathbb{E}\left[\langle\nabla F_k(\mathbf{w}_t), \mathbf{w}_{t,i}^k - \mathbf{w}_t\rangle\right]
$$

$$
\geq -\frac{1}{2N}\sum_{k=1}^{N}\sum_{i=1}^{\tau}\mathbb{E}\left[\eta_t\|\nabla F_k(\mathbf{w}_t)\|^2 + \frac{1}{\eta_t}\|\mathbf{w}_{t,i}^k - \mathbf{w}_t\|^2\right]
$$

$$
\geq -\frac{1}{N}\sum_{k=1}^{N}\sum_{i=1}^{\tau}\mathbb{E}\left[\eta_t L\left[F_k(\mathbf{w}_t) - F_k^*\right] + \frac{1}{2\eta_t}\|\mathbf{w}_{t,i}^k - \mathbf{w}_t\|^2\right]. \tag{32}
$$

Where the first inequality is by the convexity of $F_k(\cdot)$, the second inequality is by Cauchy-Schwarz inequality, and the third inequality is by the $L$-smoothness of $F_k(\cdot)$, $\|\nabla F_k(\mathbf{w}_t)\|^2 \leq 2L(F_k(\mathbf{w}_t) - F_k^*)$.

According to the above formula, we can obtain the bounds of $-\frac{\gamma_t}{N}\sum_{k=1}^{N}\sum_{i=1}^{\tau}\mathbb{E}\left[F_k(\mathbf{w}_{t,i}^k) - F^*\right]$

$$
-\frac{\gamma_t}{N}\sum_{k=1}^{N}\sum_{i=1}^{\tau}\mathbb{E}\left[F_k(\mathbf{w}_{t,i}^k) - F^*\right]
$$
$$
\leq \frac{\gamma_t}{N}\sum_{k=1}^{N}\sum_{i=1}^{\tau}\mathbb{E}\left[\eta_t L\left(F_k(\mathbf{w}_t) - F_k^*\right) + \frac{1}{2\eta_t}\|\mathbf{w}_{t,i}^k - \mathbf{w}_t\|^2 - (F(\mathbf{w}_t) - F^*)\right]. \tag{33}
$$

For $D_2$, recall the property of $\gamma_t$ in Eq. (31), $0 \leq \gamma_t \leq 2\eta_t$, substituting Eq. (33) into Eq. (31), we have

$$
-\frac{\gamma_t}{N}\sum_{k=1}^{N}\sum_{i=1}^{\tau}\mathbb{E}\left[F_k(\mathbf{w}_{t,i}^k) - F^*\right] + 2L\eta_t^2\tau(\tau+1)\Gamma
$$
$$
\leq \frac{\gamma_t}{N}\sum_{k=1}^{N}\sum_{i=1}^{\tau}\mathbb{E}\left[\eta_t L\left(F_k(\mathbf{w}_t) - F_k^*\right) + \frac{1}{2\eta_t}\|\mathbf{w}_{t,i}^k - \mathbf{w}_t\|^2 - (F(\mathbf{w}_t) - F^*)\right] + 2L\eta_t^2\tau(\tau+1)\Gamma
$$
$$
= \frac{\gamma_t(\eta_t L - 1)}{N}\sum_{k=1}^{N}\sum_{i=1}^{\tau}\mathbb{E}\left[F(\mathbf{w}_t) - F^*\right] + (2L\eta_t^2\tau^2 + 2L\eta_t^2\tau + \gamma_t\eta_t L\tau)\Gamma
$$
$$
+ \frac{\gamma_t}{2N\eta_t}\sum_{k=1}^{N}\sum_{i=1}^{\tau}\mathbb{E}\left[\|\mathbf{w}_{t,i}^k - \mathbf{w}_t\|^2\right]
$$
$$
\leq (2L\eta_t^2\tau^2 + 4L\eta_t^2\tau)\Gamma + \frac{1}{N}\sum_{k=1}^{N}\sum_{i=1}^{\tau}\mathbb{E}\left[\|\mathbf{w}_{t,i}^k - \mathbf{w}_t\|^2\right]. \tag{34}
$$

So, for $\mathbb{E}\left[\|\mathbf{v}_{t+1} - \mathbf{w}^*\|_2^2\right]$, substituting Eq. (34) and Eq. (30) into Eq. (29), we have

$$
\mathbb{E}\left[\|\mathbf{v}_{t+1} - \mathbf{w}^*\|_2^2\right] \leq (1 - \mu\eta_t\tau(1-\eta_t))\mathbb{E}\left[\|\mathbf{w}_t - \mathbf{w}^*\|_2^2\right]
$$
$$
+ (2L\eta_t^2\tau^2 + 4L\eta_t^2\tau)\Gamma + \underbrace{\frac{(2+\mu(1-\eta_t))}{N}\sum_{k=1}^{N}\sum_{i=1}^{\tau}\mathbb{E}\left[\|\mathbf{w}_{t,i}^k - \mathbf{w}_t\|_2^2\right]}_{F}. \tag{35}
$$

For $F$, according to the fact $\mathbf{w}_{t,i}^k - \mathbf{w}_t = \sum_{j=1}^{i}\eta_t\nabla F_k\left(\mathbf{w}_{t,j}^k, \xi_{t,j}^k\right)$, and Assumption 3.4.3, the expected squared $l_2$-norm of the stochastic gradients is bounded. We have

$$
\frac{(2+\mu(1-\eta_t))\eta_t^2}{N}\sum_{k=1}^{N}\sum_{i=1}^{\tau}\mathbb{E}\left[\left\|\sum_{j=1}^{i}\nabla F_k\left(\mathbf{w}_{t,j}^k, \xi_{t,j}^k\right)\right\|_2^2\right] \leq (2+\mu-\mu\eta_t)\eta_t^2 G^2\frac{\tau(\tau+1)(2\tau+1)}{6}. \tag{36}
$$

So, for the upper bound of $\mathbb{E}\left[\|\mathbf{v}_{t+1} - \mathbf{w}^*\|_2^2\right]$, due to $1 - \eta_t < 1$, substituting Eq. (36) into Eq. (35), we have

$$
\mathbb{E}\left[\|\mathbf{v}_{t+1} - \mathbf{w}^*\|_2^2\right] \leq (1 - \mu\eta_t\tau(1-\eta_t))\mathbb{E}\left[\|\mathbf{w}_t - \mathbf{w}^*\|_2^2\right]
$$
$$
+ (2+\mu(1-\eta_t))\eta_t^2 G^2\frac{\tau(\tau+1)(2\tau+1)}{6} + (2L\eta_t^2\tau^2 + 4L\eta_t^2\tau)\Gamma
$$
$$
\leq (1 - \mu\eta_t\tau(1-\eta_t))\mathbb{E}\left[\|\mathbf{w}_t - \mathbf{w}^*\|_2^2\right]
$$
$$
+ (2+\mu)\eta_t^2 G^2\frac{\tau(\tau+1)(2\tau+1)}{6} + (2L\eta_t^2\tau^2 + 4L\eta_t^2\tau)\Gamma. \tag{37}
$$

## C.9 PROOFS OF THEOREM C.3.1

According to Lemma C.2.1 to C.2.3, and a learning rate $\eta_t$ such that $0 < \eta_t \le \min\{\frac{1}{\mu B}, \frac{1}{L(\tau+1)}\}$, it can be concluded:

$$
\begin{aligned}
\mathbb{E}\left[\|\mathbf{w}_{t+1} - \mathbf{w}^*\|_2^2\right] \le &(1 - \mu\eta_t\tau(1-\eta_t))\mathbb{E}\left[\|\mathbf{w}_t - \mathbf{w}^*\|_2^2\right] + \frac{2\sum_{r=R+1}^{D}\sigma_r^2}{M\sum_{r=1}^{D}\sigma_r^2}\eta_t^2\tau^2 G^2 \\
&+ (2+\mu)\,\eta_t^2 G^2 \frac{\tau(\tau+1)(2\tau+1)}{6} + (2L\eta_t^2\tau^2 + 4L\eta_t^2\tau)\Gamma + \frac{2\eta_t^2\tau^2 G^2}{M} \\
\le &(1 - \mu\eta_t B)\mathbb{E}\left[\|\mathbf{w}_t - \mathbf{w}^*\|_2^2\right] + \eta_t^2 C,
\end{aligned}
\tag{38}
$$

where

$$
B = \tau - \frac{\tau}{L(\tau+1)}, C = (2+\mu)\,G^2\frac{2\tau^3 + 3\tau^2 + \tau}{6} + (2L\tau^2 + 4L\tau)\Gamma + \frac{2(1+\rho_R)\tau^2 G^2}{M}. \tag{39}
$$

## C.10 PROOFS OF COROLLARIES

### C.10.1 PROOF OF COROLLARY C.4.1

Assuming that Assumptions 3.4.1 to 3.4.4 hold with $\mu \ge 0$, we consider a constant learning rate $\eta$ such that $0 < \eta \le \min\left\{\frac{1}{\mu B}, \frac{1}{L(\tau+1)}\right\}$. According to Theorem C.3.1, we have

$$
\mathbb{E}\left[\|\mathbf{w}_T - \mathbf{w}^*\|_2^2\right] \le (1 - \mu\eta B)^T \mathbb{E}\left[\|\mathbf{w}_0 - \mathbf{w}^*\|_2^2\right] + \sum_{t=1}^{T}\eta^2(1-\mu\eta B)^{T-t}C. \tag{40}
$$

From the $L$-smoothness of function $F(\cdot)$, $\mathbb{E}[F(\mathbf{w}_T)] - F^* \le \frac{L}{2}\mathbb{E}\left[\|\mathbf{w}_T - \mathbf{w}^*\|_2^2\right]$, after $T$ global iterations, we have

$$
\mathbb{E}[F(\mathbf{w}_T)] - F^* \le \frac{L}{2}(1 - \mu\eta B)^T \mathbb{E}\left[\|\mathbf{w}_0 - \mathbf{w}^*\|_2^2\right] + \frac{L}{2}\sum_{t=1}^{T}\eta^2(1-\mu\eta B)^{T-t}C. \tag{41}
$$

### C.10.2 PROOF OF COROLLARY C.4.2

Let $\Delta_t = \mathbb{E}\left[\|\mathbf{w}_t - \mathbf{w}^*\|_2^2\right]$ and consider a diminishing learning rate, $\eta_t = \frac{\beta}{t+\gamma}$ for some $\beta > \frac{1}{\mu B}$ and $\gamma > 0$ such that $\eta_1 \le \min\{\frac{1}{\mu B}, \frac{1}{L(\tau+1)}\} = \frac{1}{L(\tau+1)}$. We will prove $\Delta_t \le \frac{v}{\gamma+t}$ where $v = \max\left\{\frac{\beta^2 C}{\beta\mu B - 1}, (\gamma+1)\Delta_1\right\}$. We prove it by induction. The definition of $v$ ensures that it holds for $t = 1$. Assuming that it also holds for $\Delta_t$, we draw the conclusion

$$
\begin{aligned}
\Delta_{t+1} \le (1 - \eta_t\mu B)\Delta_t + \eta_t^2 C &\le \left(1 - \frac{\beta\mu B}{t+\gamma}\right)\frac{v}{t+\gamma} + \frac{\beta^2 C}{(t+\gamma)^2} \\
&= \frac{t+\gamma-1}{(t+\gamma)^2}v + \left[\frac{\beta^2 C}{(t+\gamma)^2} - \frac{\beta\mu B - 1}{(t+\gamma)^2}v\right] \le \frac{v}{t+\gamma+1}.
\end{aligned}
\tag{42}
$$

Then by the $L$-smoothness of $F(\cdot)$, we have $\mathbb{E}[F(\mathbf{w}_t)] - F^* \le \frac{L}{2}\Delta_t \le \frac{L}{2}\frac{v}{\gamma+t}$. We choose $\beta = \frac{2}{\mu B}$, $\gamma = \frac{2L^2(\tau+1)^2}{\mu\tau(L(\tau+1)-1)} - 1$, and denote $\kappa = \frac{L}{\mu}$. Therefore, $\eta_t$ can be further expressed as $\eta_t = \frac{2}{\mu B(\gamma+t)}$. we have

$$
\nu = \max\left\{\frac{\beta^2 C}{\beta\mu B - 1}, (\gamma+1)\Delta_1\right\} \le \frac{\beta^2 C}{\beta\mu B - 1} + (\gamma+1)\Delta_1 = \frac{4C}{\mu^2 B^2} + (\gamma+1)\Delta_1, \tag{43}
$$

and

$$
\mathbb{E}[F(\mathbf{w}_T)] - F^* \le \frac{L}{2}\frac{v}{\gamma+t} \le \frac{\kappa}{\gamma+t}\left(\frac{2C}{\mu B^2} + \frac{\mu(\gamma+1)}{2}\Delta_1\right). \tag{44}
$$

## D  Hyperparameters used in baseline algorithms

Besides the hyperparameter setting provided in the main body, the other hyperparameters are as follows: For FedProx, we set $\mu = 0.01$; for MOON, we set $\tau = 1, \mu = 0.01$; for FedGen, the server epoch is 1000 and the generator learning rate is 0.005; for FedDC, we set $\alpha = 0.5$; for FedDyn, we set $\alpha = 0.5$; for FedNTD, we set $\beta = 0.001, \tau = 1$. For communication-efficient algorithms, we set $\delta = 0.05$ in LBGM; for signSGD and STC, we set their compression ratios as 1/32. Besides, We use the SGD optimizer in all experiments with momentum set to 0.

## E  Further discussion

We found that the global model space of federated learning has low rank properties. In fact, due to the scarcity of client data, federated learning algorithms face the risk of overfitting. By restricting the local model to a low dimensional subspace, the degree of freedom in model updates is reduced. This can be used as a basis for many federated learning algorithms to improve their generalization capabilities.

In addition, since the global model update of federated learning can be represented with fewer orthogonal bases, `FLSS` can also be widely integrated as a compression strategy into various compression frameworks to further reduce the compression rate. Compared with traditional compression schemes, `FLSS` pays more attention to the distribution of model parameters or gradient space. `FLSS` can adaptively select appropriate orthogonal bases to represent model updates for different networks and different scenarios. In other words, the compression of `FLSS` is data-driven and task-relevant.

In fact, many algorithms address the heterogeneity problem by considering local and global consistency. For instance, model parameter consistency is tackled by FedProx, representation consistency by Moon, and logit consistency by FedNTD. In contrast to these approaches, our method emphasizes the directional consistency between global and local updates, constraining the update direction by applying a projection to limit the angle.

