# OpenReview forum: "Federated Learning in Streaming Subspace"
_ICLR.cc/2025/Conference — Submitted to ICLR 2025_

### Official Review · Reviewer_5kEw · 2024-10-29

**Soundness:** 3
**Presentation:** 3
**Contribution:** 3
**Rating:** 5
**Confidence:** 4

**Summary:**

The paper addresses key challenges in Federated Learning (FL), particularly communication inefficiency and performance degradation with non-IID data distributions, and presents a novel solution with Federated Learning in Streaming Subspace (FLSS). The authors introduce the streaming subspace update strategy that constrains local model updates to a low-dimensional subspace aligned with the global model trajectory. By restricting updates within this subspace, the proposed FLSS framework achieves substantial communication savings without sacrificing model accuracy.

**Strengths:**

1. The authors’ approach of leveraging low-rank properties of global model trajectories is both interesting and promising, addressing the high-dimensional challenges in FL. The streaming subspace strategy is well-motivated and technically sound, demonstrating a clear understanding of how to exploit these properties for better communication efficiency and performance.

2. The methodological design of FLSS is carefully constructed, with a focus on both performance improvement and communication efficiency. The analysis indicates that the subspace update effectively handles non-IID data, addressing prevalent FL challenges that often hinder real-world deployment.

**Weaknesses:**

There are two primary concerns with the reported experiments. First, the accuracy improvements of FLSS over FedAvg and other baselines—without parameter compression—seem unexpected for a compression-based method, raising questions about the comparative setup. Normally, compression would loss information, leading to performance degradataion.

Second, some results are lower than expected; for instance, the CIFAR10 accuracy falls below 0.7, while the original FedAvg paper can report around 0.8 for CIFAR10, not accounting for the advances in FL algorithms since then.

**Questions:**

I wonder if the algorithm applies for flexible client participation where the clients are not uniformly sampled. Understanding its adaptability in such settings would clarify its applicability in dynamic environments.

---

### Official Review · Reviewer_M5kF · 2024-11-02

**Soundness:** 2
**Presentation:** 2
**Contribution:** 2
**Rating:** 3
**Confidence:** 4

**Summary:**

This work introduces FLSS, a method that maps local updates into a common low-dimensional subspace, to reduce communication overhead and align local and global updates in FL. They provide both theoretical analysis and empirical evaluations to validate their method.

**Strengths:**

1. The paper presents a sound motivation. The central idea of constraining local updates within a common low-dimensional subspace is reasonable.

2. Comprehensive experiments are provided, especially the results in Section B.1. These results effectively illustrate the differences between projecting gradients vs model updates, an interesting finding that deserves further investigation.

**Weaknesses:**

**Problem Formulation and Notation**

1. The term $\tilde{g}_t$ is introduced without a clear definition. On line 164, it is described as the projection coefficient of the negative gradient in the low-dimensional space. However, shouldn't $\text{Proj}_P(g)$ represent the projection of the gradient onto a subspace? It is unclear why the projection is taken over $\tilde{g}_t$, which is already a projection in a subspace. Please provide a precise definition of $\tilde{g}_t$ and clarify the projection operations involved.

2. In eq. (3), the variable $z$ is introduced without clear context. $w$ is defined in terms of $z$ and $z$ in terms of $w$, leading to a circular definition.

3. Figure 3(a) is identical to a figure in Li et al.'s work [1]. Please ensure proper citations are included in the figure caption.

**Algorithm**

1. In step 9 of Algorithm 1, the condition for transmitting the projected model update vector is stated as "Transmit ... according to $\text{mod}(t - L, s)$." This phrasing is vague. From my understanding, the transmission occurs when $\text{mod}(t - L, s) \neq 0$. Is this correct? A clear and explicit condition for when transmissions occur is needed, and I highly recommend rewriting Algorithm 1 for clarity. Additionally, the subspace updating method (Algorithm 2) should be introduced before Algorithm 1.

2. In step 9, the operation $\text{Proj}_{P^T}$ is unclear: why it is the transpose? Since the individual $\tilde{g}$ in step 9 and the aggregated $\tilde{g}_t$ in step 11 are already projections onto a subspace, it is confusing why another projection is necessary in step 6.

**Technical Contribution**

The technical contribution of this work is trivial to me.

1. The algorithm suggests that the subspace is updated dynamically. However, Section 3.4 assumes that the projection matrix $P$ is known beforehand and remains fixed during model updates. This presents an inconsistency between the algorithm and theoretical analysis.

2. Assumption 3.4.4 posits that the expectation of $g_t$ lies within a subspace for all $t$, which is too restrictive and unrealistic. This assumption implies that the condition holds regardless of initialization, which may not be feasible in practice.

References:

[1] Li, Tao, et al. "Low Dimensional Landscape Hypothesis Is True: DNNs Can Be Trained in Tiny Subspaces." arXiv preprint arXiv:2103.11154 (2021).

[2] Li, Xiang, et al. "On the Convergence of FedAvg on Non-IID Data."arXiv preprint arXiv:1907.02189 (2019).

**Questions:**

See the above "Weakness" section.

---

### Official Review · Reviewer_ewwP · 2024-11-08

**Soundness:** 3
**Presentation:** 2
**Contribution:** 3
**Rating:** 5
**Confidence:** 4

**Summary:**

This paper introduces Federated Learning in Streaming Subspace (FLSS), a method that constrains client model updates to a low-dimensional subspace informed by the trajectory of the global model across clients. This approach reduces communication costs and addresses the challenges of heterogeneous clients in federated learning. To identify an effective low-dimensional subspace and ensure that the global optimum lies within it, FLSS periodically samples the full, non-compressed global model updates and uses a streaming subspace tracking algorithm to adapt the subspace dynamically during training.

This paper shows convergence when each client has a strongly convex objective based on the error due to projecting client model updates to a fixed global subspace. Additionally, FLSS is compared comprehensively with existing federated learning baselines on several vision and language datasets in classification tasks, considering both model performance (classification accuracy) and communication cost. Ablation studies further explore the impact of key hyperparameters, data heterogeneity, scalability, and other factors critical to FLSS.

**Strengths:**

1. The experiments are extensive, with the proposed FLSS method evaluated against many baselines across multiple datasets. The paper also includes an ablation study to assess the impact of various hyperparameters.

2. The empirical results are promising, showing that FLSS achieves strong performance while effectively addressing the challenges of data heterogeneity and maintaining low communication costs.

3. Additionally, the application of streaming subspace tracking seems to be novel in the context of federated learning.

**Weaknesses:**

1. $\textbf{The presentation could benefit from several improvements}$.

1.1. Figures 1 and 2 are difficult to read, with captions that lack sufficient detail. In particular, Figure 2 does not clarify the meaning of the x-axis and y-axis, and Figure 1 is not referenced in the text.

1.2. The notation $z_{t+1}^k - z_t$ in Eq.(3) appears confusing and may be unnecessary.

1.3. The most significant clarity issue arises in Section 3.3. It would help if this section started by explaining the purpose of periodic sampling of the full model update and why it is essential. Specifically, without periodic sampling, the underlying subspace would remain static and not adapt over time.

The explanation of subspace tracking in lines 227–240 is a bit confusing at first glance.
Clarifying that the goal is to perform SVD on $[\textbf{G}_L, \textbf{G}_S]$, where $\textbf{G}_S$ contains all subsampled full model updates after the initial $L$ rounds, without storing all the sampled model updates, would improve readability.

Also, in line 243, the paper mentions performing SVD on $[\lambda U_1 \\Sigma_1, g_t]$ to find the new subspace, upon seeing the new full model update $g_t$, where $U_1, \Sigma_1$ are the left two factors of the SVD from $\textbf{G}_L$.

However, based on Algorithm 2, the part $U_1, \Sigma_1$ should be changing, whenever a new subspace is found. Emphasizing that the goal of the streaming algorithm is to update the subspace whenever a new $g_t$ arrives, and that the new $g_t$  can be discarded once the subspace is updated, would enhance clarity.

2. $\textbf{The theoretical analysis in Section 3.4 needs improvement}$.

2.1. The convergence analysis relies on the assumption that each client has a strongly convex objective, which is not typical in federated learning; many works address cases with non-convex objectives for each client. It’s unclear why non-convex objectives analysis or even just convex objectives analysis cannot be done in the settings this work considers, which would better align with the settings commonly used in federated learning experiments. This disconnect is also apparent in the experimental setup, which does not focus on strongly convex cases.

2.2. The analysis is based on a fixed subspace. However, the proposed FLSS first performs communication of full client model updates for $L$ rounds, then samples full client model updates every $s$ round, and uses the streaming algorithm to update the subspace with a hyperparameter $\lambda$ that essentially determines the weight of the old subspace. How does $L$, $\lambda$ and $s$ affect the convergence rate then? For instance, one would expect a smaller $s$ to lead to faster convergence, yet none of these critical hyperparameters appear in the convergence bound. This omission leaves the theoretical results disconnected from the proposed algorithm FLSS.

2.3. Furthermore, the paper claims that subspace projection of model updates mitigates the effects of data heterogeneity across clients. Ideally, the theoretical analysis should compare the terms involving dissimilarity and error due to subspace tracking in the convergence bound with corresponding terms in the convergence bounds of previous works to highlight any potential theoretical improvement.

2.4. While the paper dedicates considerable effort to analyzing the effects of factors such as data heterogeneity, the number of clients, and the number of local steps on the convergence rate, these influences are already well understood. A more meaningful contribution would be to analyze the impact of the novel components introduced by FLSS, such as subspace tracking and periodic sampling, on convergence.

3. $\textbf{Practical concerns about memory and storage requirements per client need to be addressed}$.

FLSS requires each client to store the subspace used to project their model updates, which adds an additional storage demand of $D \times R$, where $D$ is the number of model parameters and $R$ is the dimension of the subspace. This is equivalent to saying each client needs to store an additional $R$ copies of their local model, which can pose a problem in federated learning, where clients are often edge devices like phones or tablets with limited storage capacity. In line 327 the paper states that such space requirement is “often negligible relative to the scale of local data”. This is not necessarily true in practice. This raises concerns about the memory / storage requirement from a client’s perspective.

4. $\textbf{Regarding the experiments}$, Figure 4 demonstrates that increasing the number of initial rounds $L$ without compressing model updates increases the model accuracy. However, this also increases total communication costs. It would be better to show this trade-off in this figure.

5. $\textbf{Minor issues}$.

5.1. Notation overload: $L$ represents both the number of initial sampling rounds and the smoothness parameter of each client’s objective.

5.2. $Proj_{P^T}(g_t^k)$ in line 11 of Algorithm 1 should be $Proj_{P}(g_t^k)$?

**Questions:**

1. Regarding the convergence bound in Theorem 3.4.1, it is common to see that the convergence rate depends on $\eta^2 \tau^2$, where $\eta$ is the learning rate and $\tau$ is the number of local steps, but it is very weird to see $\eta^2 \tau^3$ in the bound. This implies the dependency on number of local steps is significantly worse (in terms of convergence) in FLSS than it is in other FL algorithms. Where does the $\tau^3$ come from in this case? Is there any intuitive explanation?

2. What is the value of $\lambda$ in FLSS used in experiments presented in section 4.2?

3. In line 325, shouldn’t the average communication cost per round also depend on the number of initial rounds $L$?

---

### Official Review · Reviewer_S1h5 · 2024-11-09

**Soundness:** 3
**Presentation:** 3
**Contribution:** 3
**Rating:** 6
**Confidence:** 5

**Summary:**

The authors present FLSS (Federated Learning in Streaming Subspace), a framework that addresses federated learning's challenges of high communication costs and decreased performance in non-IID data contexts. By confining local updates to a low-dimensional global streaming subspace, FLSS significantly reduces communication overhead while maintaining model quality. By leveraging the low-rank properties of global model trajectories, FLSS offers a promising solution for scalable and efficient FL. Experiments on CIFAR-100 demonstrate the effectiveness of this approach, showing a 2.14% performance improvement over the baseline and an 80% reduction in communication costs

**Strengths:**

1. The authors introduce a streaming subspace update strategy, limiting local model updates to a global streaming subspace, which creates low-dimensional trajectories and reduces overall data dimensionality.
2. Using the FLSS framework, these reduced-dimensional trajectories from local models are aggregated to update the global model, allowing the server to capture only essential information and cutting down on communication costs.
3. Extensive experiments across diverse datasets reveal that the FLSS-enabled FL method not only outperforms the baseline but also significantly minimizes communication overhead.

**Weaknesses:**

1. On CIFAR-100, the performance with a beta value of 0.1 matches that of FedAvg. Why might this be? Further testing with even lower beta values (e.g., 0.01) is needed to explore performance on more complex datasets, like ImageNet or CIFAR-100.
2. Experiments with a larger client base (e.g., 100 clients) are essential to evaluate the scalability of the proposed method.
3. Subspaces are generally robust to noise. Testing on noisy-label datasets would help confirm the robustness of this approach.

**Questions:**

NA

---

### Meta-Review · Area_Chair_Qg32 · 2024-12-22

**Metareview:**

The authors present FLSS (Federated Learning in Streaming Subspace), a framework that addresses federated learning's challenges of high communication costs and decreased performance in non-IID data contexts. By confining local updates to a low-dimensional global streaming subspace, FLSS significantly reduces communication overhead while maintaining model quality. By leveraging the low-rank properties of global model trajectories, FLSS offers a promising solution for scalable and efficient FL. Experiments on CIFAR-100 demonstrate the effectiveness of this approach, showing a 2.14% performance improvement over the baseline and an 80% reduction in communication costs.

Summary of strengths:
- The authors introduce a streaming subspace update strategy, limiting local model updates to a global streaming subspace, which creates low-dimensional trajectories and reduces overall data dimensionality.
- Using the FLSS framework, these reduced-dimensional trajectories from local models are aggregated to update the global model, allowing the server to capture only essential information and cutting down on communication costs.
- Extensive experiments across diverse datasets reveal that the FLSS-enabled FL method not only outperforms the baseline but also significantly minimizes communication overhead.
- The empirical results are promising, showing that FLSS achieves strong performance while effectively addressing the challenges of data heterogeneity and maintaining low communication costs.
- Additionally, the application of streaming subspace tracking seems to be novel in the context of federated learning.

Summary of weaknesses:
- On CIFAR-100, the performance with a beta value of 0.1 matches that of FedAvg. Why? Further testing with even lower beta values (e.g., 0.01) is needed to explore performance on more complex datasets, like ImageNet or CIFAR-100.
- Experiments with a larger client base (e.g., 100 clients) are essential to evaluate the scalability of the proposed method.
- Subspaces are generally robust to noise. Testing on noisy-label datasets would help confirm the robustness of this approach.
- Issues with presentation of the results; lack of clarity in notation and algorithm part
- Issues with theory in Sect 3.4
- Concerns with experiments (e.g., memory & storage).
- Contributions seen as "trivial" by at least one reviewer

---

The authors did not write a rebuttal, which shows low confidence about their ability to handle the criticism raised. The scores for the paper were mixed (6, 5, 3, 5) -- but 3 out of 4 reviewers tended towards rejection. Perhaps a rebuttal and discussion could change the views of the reviewers, but no rebuttal was submitted. I have no other option than to recommend the paper for rejection.

**Additional Comments On Reviewer Discussion:**

No rebuttal was submitted; and there was therefore also no need for a discussion.

---

### Decision · Program_Chairs · 2025-01-22

Reject